# SEQUENCE LEARNING FROM CONTINUOUS STREAMS OF DATA

## ABSTRACT

Sequence data are inherently dependent, yet sequence learners (e.g., language models) are often trained as if samples were independent and identically distributed (IID) by segmenting long streams into short, shuffled chunks, breaking natural continuity and undermining long-range credit assignment. We formalize *multi-stream sequence learning*, a continuity-preserving training framework that presents multiple streams in their natural order, a setting that has been conflated with solution methods and remains underexplored. To support this paradigm, we propose *Memora*, a recurrent-only architecture with persistent hidden states, making it more suitable for sequence learning than architectures trained with IID chunking. Memora is built around our *Gated Linear Recurrent Unit* (GLRU), a lightweight unit designed for efficient parallel training and robust temporal reasoning. It achieves effective learning on long byte-level sequences and remains reliable even in the strict streaming setting, where data arrive online one byte at a time. Our experiments highlight that continuity-preserving training outperforms IID chunking, underscoring the importance of continuity in sequence learning.

## 1 INTRODUCTION

Modern sequence models, particularly Transformers, have achieved remarkable performance across diverse domains by adopting the independent-and-identically-distributed (IID) training paradigm, where long continuous data streams are partitioned into randomized, fixed-length segments. This segmentation strategy fully exploits modern hardware, achieves high-throughput parallel processing, and underpins state-of-the-art results in language, vision, and beyond (Brown et al. 2020, Hoffmann et al. 2022, Touvron et al. 2023, Guo et al. 2025). Yet this very convenience comes at the expense of severing the temporal continuity intrinsic to data streams. Natural streams (e.g., linguistic text, audio waveforms, videos, or genomic code) may rely on dependencies that span far beyond individual chunks. As a stopgap, practitioners extend context windows (Ding et al. 2024, Pal et al. 2023, Wang et al. 2024b), an approach that rapidly escalates computational cost and becomes impractical for those with limited resources (Huang et al. 2024). This tension invites a pivotal question: Does the IID regime inherently constrain the capacity to learn genuine long-range dependencies?

The temporal segmentation not only disrupts the natural continuity of streams but also hinders the model's ability to capture dependencies that extend beyond segment boundaries. Transformers, when trained on arbitrarily chunked windows, lack mechanisms to relate patterns across segments, leaving long-range structure unmodeled. While recurrent architectures (Hochreiter and Schmidhuber 1997, Arjovsky et al. 2016, Orvieto et al. 2023) are, in principle, better suited for such dependencies by maintaining hidden state over time, modern variants, except for a few works (e.g., Dai et al. 2019, Hutchins et al. 2022), typically conform to the same IID regime. In practice, they reset hidden states at segment boundaries (e.g., Gu and Dao 2024, De et al. 2024, Beck et al. 2024), discarding accumulated memory and undermining their temporal expressivity. A recurrent framework that fully leverages persistent state across long sequences remains largely unrealized.

Motivated by the limitations of the IID training paradigm and the untapped capacity of recurrent architectures, we formalize the *multi-stream sequence learning* paradigm, a training framework that maintains temporal coherence by presenting multiple parallel data streams in their natural order and resetting states only at sequence boundaries (e.g., end of document). Drawing on ideas from Dai et al. (2019), we formalize the problem of preserving contextual memory across segment boundaries,

enabling sequential updates that retain memory and support long-range dependency modeling that extends beyond each update block. This paradigm aligns naturally with real-world applications like online inference and real-time decision-making, including autonomous vehicles (Verma et al. 2023), video prediction (Carreira et al. 2024, Yoo et al. 2024), adaptive chatbots (Dai et al. 2025), streaming learning (Elsayed et al. 2024), and robotics (Vasan et al. 2024), where the assumption of independent samples does not hold. Under this setting, a recurrent architecture can maintain a persistent state across an entire single stream to perform long-range credit assignment, unlocking modeling capabilities that IID training inherently disrupts and revitalizing the strength of recurrence.

Realizing multi-stream sequence learning at scale requires an architecture that balances parallel training efficiency and the ability to learn online—we propose *Memora*, a lightweight recurrent-only backbone built on the *Gated Linear Recurrent Unit* (GLRU). GLRU employs gating mechanisms with a parallelizable formulation, narrowing the throughput gap with Transformers. Moreover, GLRU enables stable performance under various update strides, including a stride of 1, making Memora adaptable to diverse training scenarios, from offline pretraining to real-time learning.

Through extensive evaluations on byte-level sequence modeling, we demonstrate that Memora trained under the multi-stream paradigm consistently surpasses models trained with the IID paradigm. Our work contributes (1) a formalization of the multi-stream sequence learning paradigm, (2) the Memora architecture with the GLRU cell for efficient, scalable recurrence supporting both truncated backpropagation through-time and real-time recurrent learning, and (3) empirical evidence in offline pretraining and online learning where continuity-aware training unlocks performance gains on long-sequence tasks for recurrent-based models.[1]

## 2 BACKGROUND ON RECURRENT LEARNING

Let us consider a recurrent module with dynamics that can be written as $h_t = f(h_{t-1}, x_t, \theta)$, where $h_t \in \mathbb{R}^n$ is the hidden-state vector, $x_t \in \mathbb{R}^d$ is the input vector, and $\theta$ is a set of learnable parameters of the recurrence function, containing the input-weight and recurrence-weight matrices. The output is given by $\hat{y}_t = g(h_t, x_t, \phi)$, where $\hat{y}_t \in \mathbb{R}^m$, and $\phi$ is a set of learnable parameters of the output function. To learn the parameters, we need to compute $\nabla \mathcal{L}_W, \ \forall W \in \theta$ and $\nabla \mathcal{L}_V, \ \forall V \in \phi$. Given a target $y_t \in \mathbb{R}^m$, the gradient of the loss with respect to $W$, treating it as a flattened vector, is given by

$$\frac{\partial \mathcal{L}(y_t, \hat{y}_t)}{\partial W} = \left(\frac{\mathrm{d}h_t}{\mathrm{d}W}\right)^\top \frac{\partial \mathcal{L}}{\partial h_t}, \quad \forall W \in \theta. \tag{1}$$

### 2.1 BACKPROPAGATION THROUGH TIME (BPTT)

Note that to obtain $\frac{\mathrm{d}h_t}{\mathrm{d}W}$, we need to consider both the direct and indirect effects of changing $W$ on $h_t$ since there are direct and indirect gradient paths at each time step. Since we can unroll the function $h_t = f(f(f(\ldots f(h_1, x_0, \theta), \ldots x_{t-2}, \theta), x_{t-1}, \theta), x_t, \theta)$, we write the gradient $\frac{\mathrm{d}h_t}{\mathrm{d}W}$ as

$$\frac{\mathrm{d}h_t}{\mathrm{d}W} = \frac{\partial h_t}{\partial W} + \frac{\partial h_t}{\partial h_{t-1}}\frac{\mathrm{d}h_{t-1}}{\mathrm{d}W}, \quad \forall W \in \theta. \tag{2}$$

We need to keep unrolling further because $W$ again affects $h_{t-1}$ through two pathways. Let us define $I_t \doteq \frac{\mathrm{d}h_t}{\mathrm{d}W}$, $J_t \doteq \frac{\partial h_t}{\partial W}$, and $K_t \doteq \frac{\partial h_t}{\partial h_{t-1}}$. We can write the recursive relationship as follows:

$$I_t = J_t + K_t I_{t-1}$$
$$= J_t + K_t J_{t-1} + K_t K_{t-1} I_{t-2}$$
$$= \sum_{j=1}^{t} \left(\prod_{i=j+1}^{t} K_i\right) J_j + \left(\prod_{i=1}^{t} K_t\right) J_0$$
$$= \sum_{j=1}^{t} \left(\prod_{i=j+1}^{t} K_i\right) J_j. \qquad \text{(under the assumption that that } J_0 = 0) \tag{3}$$

---

[1]We provide a minimal easy-to-follow implementation of Memora under the IID and multi-stream settings through this Colab Notebook.

This relationship is utilized in BPTT by efficiently calculating the summation backward. Calculating this requires storing all previous inputs and states $\boldsymbol{x}_i, \boldsymbol{h}_i, \forall\{1, \ldots, t\}$. In other words, we need to backpropagate the gradient in time, starting from the current time step and going all the way back to the first time step. The computational and memory resource grows linearly with the number of steps because we need to go from the current step to the beginning of time for each update.

## 2.2 TRUNCATED-BACKPROPAGATION THROUGH TIME (T-BPTT)

We can simplify the intensive computation needed by BPTT if we truncate the backpropagation process by going back in time up to time step $t - T$, where $t > T$. This creates the Truncated-Backpropagation Through Time (T-BPTT) gradient (Williams and Zipser 1989):

$$\frac{\mathrm{d}\boldsymbol{h}_t}{\mathrm{d}\boldsymbol{W}} \approx \sum_{j=t-T}^{t} \left( \prod_{i=j+1}^{t} \frac{\partial \boldsymbol{h}_i}{\partial \boldsymbol{h}_{i-1}} \right) \frac{\partial \boldsymbol{h}_j}{\partial \boldsymbol{W}}, \forall t > T. \tag{4}$$

T-BPTT needs only to store last $T + 1$ inputs and states to approximate the gradient. Truncating the gradient drops any interactions beyond the truncation length, which may make the learner myopic.

## 2.3 REAL-TIME RECURRENT LEARNING (RTRL)

Instead of relying on the rolled-out equation (Eq. 3 or Eq. 4), we can instead compute $\boldsymbol{I}_t = \frac{\mathrm{d}\boldsymbol{h}_t}{\mathrm{d}\boldsymbol{W}}$, known also as the sensitivity matrix, incrementally using its recursive relationship. This process is known as Real-Time Recurrent Learning (RTRL) (Williams and Peng 1990) given by

$$\boldsymbol{I}_t = \boldsymbol{J}_t + \boldsymbol{K}_t \boldsymbol{I}_{t-1}. \tag{5}$$

The quantities $\boldsymbol{J}_t$ and $\boldsymbol{K}_t$ use the current input, and they can be computed easily without BPTT, requiring only storing $\boldsymbol{I}_{t-1}$ to compute $\boldsymbol{I}_t$ in an incremental fashion. RTRL computes the true gradient without any approximation, in contrast to T-BPTT, given that the parameters remain fixed.

## 2.4 PARALLELIZATION WITH RECURRENT LEARNING

Unlike Transformers, parallelization in recurrent-based networks is challenging because they have states evolving sequentially over time. Luckily, parallel scan (Blelloch 1990) can be used to re-express certain recurrent computations as an associative scan, which for many linear (in particular diagonal) recurrent updates reduces the time complexity of training from $O(L)$ to $O(\log L)$, where $L$ is the segment length, also known as the sequence length. We provide a primer on parallel scan in Appendix H and a primer on linear recurrent units in Appendix G. In practice, the truncation length is set to the segment length, and the scan operation is performed on the input sequence to produce the outputs in parallel (e.g., Gu and Dao 2024).

Although parallel scan remedies some of the challenges for a useful subclass of recurrent units (e.g., diagonal), it does not eliminate the fundamental cost of maintaining and differentiating through arbitrary state dynamics. In particular, applying parallel scan to the sensitivity recurrent equation (Eq. 5) is computationally prohibitive in general. For example, even for the diagonal units, the cost grows as $O(n^3 \log L)$ with the sensitivity matrices on the form $\boldsymbol{S} \in \mathbb{R}^{n \times n}$. In practice, parallelizing RTRL is typically limited to the batch dimension and to very short segments; nevertheless, its strictly online, per-step updates make it better suited to real-time learning than large-scale parallelization.

## 3 MULTI-STREAM SEQUENCE LEARNING

We propose *multi-stream sequence learning*, a training framework that preserves temporal continuity across update blocks by presenting multiple data streams in their natural order. Let us consider the block next-token prediction problem. At each iteration $k$, the learner receives a block of $T$ tokens $\mathbf{x}_{k:k+T-1} \doteq \{\mathbf{x}_k, \ldots, \mathbf{x}_{k+T-1}\}$ and is tasked with predicting the conditional distribution of the subsequent tokens $\mathbf{x}_{k+1:k+T} = \{\mathbf{x}_{k+1}, \ldots, \mathbf{x}_{k+T}\}$ in parallel, which can be done by minimizing the block-level negative log-likelihood $\mathcal{L}_k(\theta) \doteq -\sum_{i=1}^{T} \log p_\theta(\mathbf{x}_{k+i} \mid \mathbf{x}_{k:k+i-1})$, where $\theta$ is the set of learnable parameters. In practice, we also process $B$ of such blocks in parallel (the batch or stream

dimension), but we drop their index in this notation for simplicity. Let $N$ denote the total dataset length. In conventional IID training, the start index $k$ is sampled uniformly from $\{1, T+1, \ldots, N-T\}$, so blocks begin at random positions. In contrast, preserving the original stream order instead requires using sequential starts $k = 1, 1+T, 1+2T, \ldots$ up to the largest start index $\leq N-T$. More generally, one may use a rolling window with stride $S \leq T$, i.e. $k = 1, 1+S, 1+2S, \ldots$, up to the largest start index $\leq N-T$, where the non-overlapping case is $S = T$. Unlike IID training, which breaks temporal continuity by randomizing block starts, this multi-stream scheme treats each stream temporally in a continuous manner and only presents a new sequence at semantically meaningful boundaries (e.g., end of document or video).

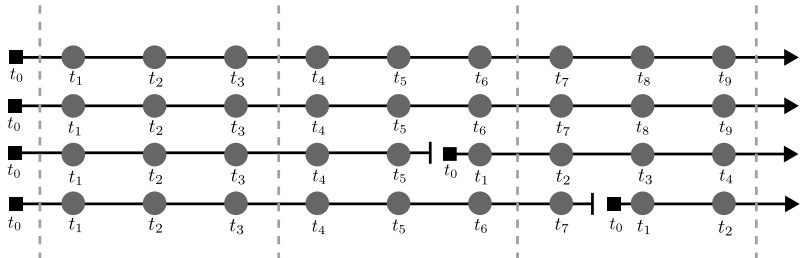

Figure 1: Overview of multi-stream sequence learning. Four streams ($B{=}4$) each yield blocks of length $S{=}T{=}3$; when an episode ends, it is immediately replaced by a new one starting at $t_0$.

Figure 1 illustrates the workflow of the multi-stream sequence learning. The learner is presented with $B$ parallel streams, each containing a series of episodes (e.g., an article, a video, or a complete conversation). When one episode ends, we dynamically swap it with a new episode that starts from $t_0$. Training is done by collecting a block of stride $S = T$ from each stream, giving an effective batch size of $B \times T$ for an update. This setup generalizes several regimes, for example, when $B = 1$ and $T = 1$, the model reduces to what is known as streaming learning (e.g., Elsayed et al. 2024). We refer to $S$ as the update stride, which is known as the sequence length for IID training. However, we reserve the latter for IID training, where—unlike in multi-stream training—the sequence is broken at the block boundary. The formulation of this framework is inspired by the ideas in Dai et al. (2019).

While each episode is randomly selected from the corpus, the temporal coherence within each episode, which may span an entire book, is preserved. This enables the model to maintain and evolve hidden states across update blocks, supporting long-range credit assignment and structured memory accumulation. Recurrent architectures naturally lend themselves to this paradigm, in contrast to Transformer-based models. Yet, current recurrent learning paradigms do not evolve hidden states across update blocks, as they pick the blocks IID. In contrast, update blocks in multi-stream learning are contiguous and from the same episode unless the episode ends before a block completes.

Multi-stream sequence learning aligns closely with real-world deployment scenarios, such as online prediction (Carreira et al. 2024), continual learning (Elsayed and Mahmood 2024), and real-time control (Vasan et al. 2024), where the assumption of independent samples breaks down. Furthermore, it retains compatibility with modern hardware by enabling efficient parallelism across streams, while unlocking new capabilities through temporal continuity.

## 4 THE MEMORA ARCHITECTURE

To study the multi-stream learning framework, we introduce the Memora architecture, a simplified recurrent-only architecture that is compatible with the multi-stream learning paradigm. The Memora architecture depends on the Gated Linear Recurrent Unit (GLRU), which is a gated recurrent architecture. We start by describing GLRU and explaining how it can be used with T-BPTT and RTRL; then, we describe the components of the Memora architectural design.

### 4.1 LEARNING TEMPORAL STRUCTURES WITH GATED LINEAR RECURRENT UNIT (GLRU)

Recent advancements in recurrent learning (e.g., Gu and Dao 2024, De et al. 2024) demonstrated the importance of gated recurrence in solving complex tasks such as language. In addition, gated recurrence has been shown to be able to implement the linear attention (Katharopoulos et al. 2020) operation (Zucchet et al. 2024, Huang et al. 2023, Dao and Gu 2024). Our design builds on previous

works and relies on gating both the input and the state. Specifically, GLRU builds on LRU (Orvieto et al. 2023), which uses complex-valued, non-gated recurrence, but incorporates a real-valued, gated recurrence instead. The GLRU recurrence formulation is given by

$$\boldsymbol{h}_t = \boldsymbol{r}(\boldsymbol{x}_t) \circ \boldsymbol{h}_{t-1} + \boldsymbol{\gamma}_t \circ \boldsymbol{g}(\boldsymbol{x}_t) \circ (\boldsymbol{B}\boldsymbol{x}_t), \tag{6}$$

where $\boldsymbol{\gamma}_t = \sqrt{1 - \boldsymbol{r}_t^2}$, $\boldsymbol{g}(\boldsymbol{x}_t) = \boldsymbol{G}\boldsymbol{x}_t$, $\boldsymbol{r}(\boldsymbol{x}_t) = e^{-ce^{\boldsymbol{\nu}} \circ \sigma(\boldsymbol{R}\boldsymbol{x}_t)}$, $\boldsymbol{x}_t \in \mathbb{R}^d$, $\boldsymbol{h}_t \in \mathbb{R}^n$, and $\boldsymbol{G}, \boldsymbol{B} \in \mathbb{R}^{n \times d}$. We set $c$ to 3. We do not apply gating on the output since the Memora architecture provides that gating, similar to the Hawk architecture (De et al. 2024). The output of the recurrence is given by $\boldsymbol{y}_t = \boldsymbol{h}_t$. Learning with T-BPTT is straightforward and can be done using Algorithm 1. However, learning with RTRL (see Algorithm 2) requires deriving the sensitivity matrices for each learnable parameter in the recurrence equation, namely: $\boldsymbol{\nu}, \boldsymbol{B}, \boldsymbol{R}, \boldsymbol{G}$. We provide the update equations here and defer the full derivation to Appendix E. The RTRL sensitivity update equations are given by

$$\mathbf{S}_t^{\boldsymbol{\nu}} = \boldsymbol{r}(\boldsymbol{x}_t) \circ \mathbf{S}_{t-1}^{\boldsymbol{\nu}} - ce^{\boldsymbol{\nu}} \circ \sigma(\boldsymbol{R}\boldsymbol{x}_t) \circ \boldsymbol{r}(\boldsymbol{x}_t) \circ \boldsymbol{h}_{t-1} + c\frac{\boldsymbol{r}(\boldsymbol{x}_t)^2}{\boldsymbol{\gamma}} \circ e^{\boldsymbol{\nu}} \circ \sigma(\boldsymbol{R}\boldsymbol{x}_t) \circ \boldsymbol{g}(\boldsymbol{x}_t) \circ (\boldsymbol{B}\boldsymbol{x}_t),$$

$$\mathbf{S}_t^{\boldsymbol{B}} = \mathrm{Diag}\left(\boldsymbol{r}(\boldsymbol{x}_t)\right) \mathbf{S}_{t-1}^{\boldsymbol{B}} + (\boldsymbol{\gamma} \circ \boldsymbol{g}(\boldsymbol{x}_t))\boldsymbol{x}_t^{\top},$$

$$\mathbf{S}_t^{\boldsymbol{G}} = \mathrm{Diag}\left(\boldsymbol{r}(\boldsymbol{x}_t)\right) \mathbf{S}_{t-1}^{\boldsymbol{G}} + (\boldsymbol{\gamma} \circ (\boldsymbol{B}\boldsymbol{x}_t))\boldsymbol{x}_t^{\top},$$

$$\mathbf{S}_t^{\boldsymbol{R}} = \mathrm{Diag}(\boldsymbol{r}(\boldsymbol{x}_t))\boldsymbol{S}_r^{\boldsymbol{R}} + \left(\boldsymbol{d} \circ \left(\boldsymbol{h}_{t-1} - \frac{\boldsymbol{r}(\boldsymbol{x}_t)}{\boldsymbol{\gamma}} \circ \boldsymbol{g}(\boldsymbol{x}_t) \circ (\boldsymbol{B}\boldsymbol{x}_t)\right)\right) \boldsymbol{x}_t^{\top},$$

where $\boldsymbol{d} = c\boldsymbol{r}(\boldsymbol{x}_t) \circ e^{\boldsymbol{\nu}} \circ \sigma(\boldsymbol{R}\boldsymbol{x}_t) \circ (1 - \sigma'(\boldsymbol{R}\boldsymbol{x}_t))$, and the division is performed element-wise.

Learning with RTRL allows learning in real-time from the samples as they arrive. It is compatible with our multi-stream learning paradigm and can achieve efficient learning with $T = 1$. One fundamental limitation of RTRL is that its parallelization with parallel scan (when $T > 1$) is expensive since the sensitivity equations are based on matrices. Therefore, we limit the usage to the case where $T = 1$ with $B$ parallel streams.

Finally, we place GLRU in the landscape of recurrent units in Table 1. Notably, our GLRU allows for state expansion since the input $\boldsymbol{x}_t$ can be expanded to a larger space using $\boldsymbol{B}$, similar to the LRU unit and unlike RG-LRU. Additionally, our design allows for efficient RTRL implementation, which is facilitated by the element-wise input and output gating. RG-LRU design, in principle, also allows for efficient RTRL. In contrast, the RTRL mode of GRU (Chung et al. 2014) is intractable and requires computational complexity of $O(n^4)$. The RTRL mode of RG-LRU is not introduced in the literature, so we derive its RTRL mode in Appendix E to compare it with GLRU in the experiments and skip comparing with the RTRL mode of Mamba due to its derivation and implementation complexity.

|  | Gated LRU (Ours) | RG-LRU (De et al. 2024) | LRU (Orvieto et al. 2023) | GRU (Chung et al. 2014) |
|---|---|---|---|---|
| **State expansion** | Yes | No | Yes | Yes |
| **RTRL mode** | Efficient | Efficient | Efficient | Intractable |
| **Gated/Selective** | Yes | Yes | No | Yes |
| **T-BPTT Scan** | Efficient | Efficient | Efficient | Intractable |

Table 1: Comparison of Gated LRU, RG-LRU, LRU, and GRU. Gated LRU is the first gated recurrent unit that supports different modes of training: efficient parallelization with T-BPTT, and fast real-time learning with RTRL.

## 4.2 ARCHITECTURAL DESIGN

Memora is a recurrent-only architecture with two main components: 1) a residual normalized gated recurrent block, followed by 2) a residual normalized gated MLP block. This design follows the general Transformer architecture outline, where the first block captures the temporal relations and the other learns representations (see Touvron et al. 2023). We apply pre-normalization using RM-SNorm (Zhang and Sennrich 2019) on each block with learnable parameters. Each block employs the gated linear unit design (Shazeer 2020), where the input is expanded then contracted using the block linear maps. This design is common across many architectures (e.g., De et al. 2024, Gu and Dao (2024)). We propose a new variation where we use layer norm (Ba et al. 2016) without learnable parameters after each linear mapping and also after each element-wise multiplication. Our

gated recurrent block is very close to the gated MLP, and instead of using a linear map in one of the GLU branches, we replace it with GLRU. Lastly, the activation $\sigma$ we use is GeLU (Hendrycks and Gimpel 2016). Figure 2 outlines the Memora architecture and its two main components.

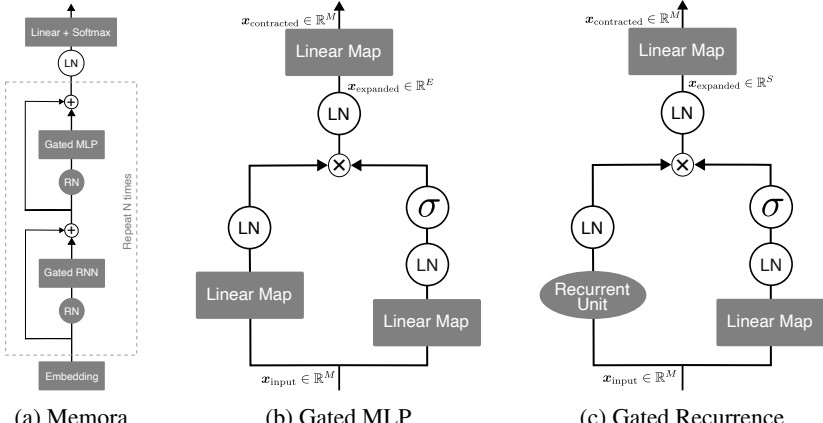

(a) Memora          (b) Gated MLP          (c) Gated Recurrence

Figure 2: The Memora architecture. Filled blocks represent components with learnable parameters. LN/RN denotes LayerNorm/RMSNorm. $M$ is the model dimension and $S$ is the state dimension.

## 5 EXPERIMENTS

In this section, we start by studying long-range memory capabilities and selective attention under high memory demand using the selective copying task (Gu and Dao 2024). Then we study the effect of varying sequence lengths for IID or update strides for multi-stream on the quality of byte-level language modeling with TinyStories (Eldan and Li 2023). After that, we study how Memora scales with the number of parameters using byte-level language modeling with FineWebEdu (Penedo et al. 2024) and DNA modeling with the human genome dataset (HG38, Schneider et al. 2017). We then study learning from one sample at a time with RTRL and 1-step BPTT. Finally, we consider GLRU alternatives to show the performance gain with GLRU when used with Memora compared to other baselines. Here, we focus on the key results and experimental details and provide the full experimental details and configurations in Appendix F.

### 5.1 SELECTIVE COPYING

The selective copying task is a variation of the original copying task (Arjovsky et al. 2016), where the learner must memorize tokens at varying positions within a sequence. This task demands context-aware reasoning and effectively differentiates models that use gating mechanisms from those that do not. Gating enables models to selectively retain or discard information, making this task easier to solve. Our experimental setup follows Gu and Dao (2024) but with a reduced training budget of $50,000$ iterations instead of $200,000$, allowing us to study training efficiency under limited compute. For simplicity, we let each model see the full episode—full BPTT is used instead of T-BPTT.

| Model | Arch. Type | Accuracy | Parameter Count |
|---|---|---|---|
| Llama2 | Transformer | $98.5200\% \pm 0.8701$ | $888,768$ |
| Hawk | Recurrent + Temp. Conv. | $99.0120\% \pm 0.4391$ | $438,080$ |
| Mamba2 | Recurrent + Temp. Conv. | $99.4280\% \pm 0.3646$ | $553,828$ |
| Memora w/ MinGRU | Recurrent-only | $53.8280\% \pm 12.5294$ | $206,080$ |
| Memora w/ LRU | Recurrent-only | $11.0160\% \pm 0.6581$ | $304,896$ |
| Memora w/ GLRU | Recurrent-only | $99.5620\% \pm 0.1918$ | $239,360$ |

Table 2: Validation Accuracy on the Selective Copying Task. Each model is trained for 50K iterations. Each episode has a length of $4096$ with only 16 numbers to remember with varying positions. The results are averaged over 5 independent runs, and we show the standard error.

Table 2 reports validation performance for Llama2 Transformer, Hawk, Mamba2 models, and Memora variants, each with two layers. Both Memora with GLRU and Hawk quickly solve the task with relatively few parameters, using model dimensionality of $64$ and state dimensionality of $256$. In contrast, Transformer and Mamba require increased model dimensionality to achieve comparable results within the same number of iterations. Notably, Memora with GLRU attains the highest accuracy with the fewest parameters, relying solely on its recurrent architecture without temporal convolutions. Lastly, replacing GLRU with another gated unit (MinGRU) or a non-gated unit (LRU) leads to a significant performance drop, underscoring the superior memory capacity of GLRU.

## 5.2 BYTE-LEVEL MODELING WITH VARYING SEQUENCE LENGTHS OR UPDATE STRIDES

We study byte-level language modeling on the TinyStories dataset (Eldan and Li 2023), focusing on how performance scales with sequence length for IID and with update stride for multi-stream, while keeping the effective batch size $BT$ fixed. In the multi-stream setup, sequences are presented in natural order, which we hypothesize favors recurrent models. In contrast, we do not expect Transformers to benefit from this structure due to their lack of persistent memory, but we still add multi-stream Transformer (Llama2) to the experiment.

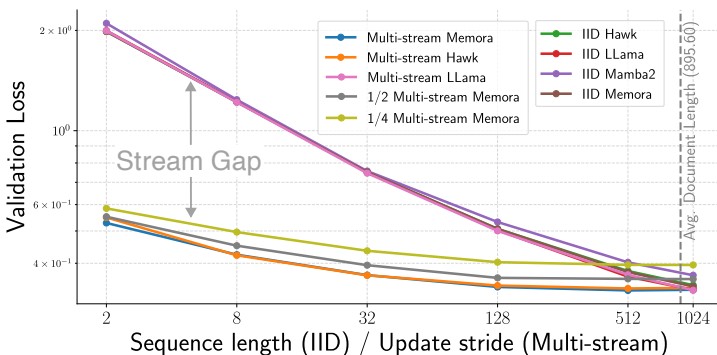

Figure 3: Performance scaling with sequence length (for the IID setting) or update stride (for the multi-stream setting). We show the final performance of different baselines trained on TinyStories using the IID and multi-stream settings. The average document size is $895.60$, which means models with sequence lengths of $1024$ may contain an entire document in an update block.

Figure 3 shows the performance of Mamba2, Llama2, Hawk, and Memora in the IID setting, and compares them with Memora in the multi-stream setting. We also adapt Hawk and Llama2 to the multi-stream setup, representing existing recurrent-based and Transformer-based models, respectively. Sequence length (or update stride) is varied from 2 to $1024$. In the IID setting, all models perform similarly but degrade significantly at shorter sequence lengths—as expected, because shorter contexts reduce temporal credit assignment. In contrast, both Memora and Hawk perform well in the multi-stream setting even with an update stride of 2. Note that multi-stream Llama2 showed no advantage over IID training due to the lack of states. This suggests that maintaining a persistent state across update blocks helps recurrent models recover long-range dependencies lost in the IID setting. We further evaluate Memora in the multi-stream setting with reduced training iterations (half and quarter). Even with limited training, Memora consistently outperforms IID-trained models across nearly all sequence lengths, further emphasizing its superior memory capabilities in multi-stream.

## 5.3 DNA MODELING

DNA sequences are naturally long, presenting a challenge for modern sequence models. For instance, human chromosome 2 contains approximately $250$ million base pairs (Hillier et al. 2005). We use the human genome dataset (HG38; Schneider et al. 2017) with the train-validation splits defined by Avsec et al. (2021). The sequences span up to $131,072$ base pairs, making this a strong benchmark for evaluating long-range dependency modeling.

In Figure 4a, we compare the performance of Llama2, Hawk, and Mamba2 in the IID setting, and contrast them with Memora and Hawk in the multi-stream setting. In IID training, Llama2 outper-

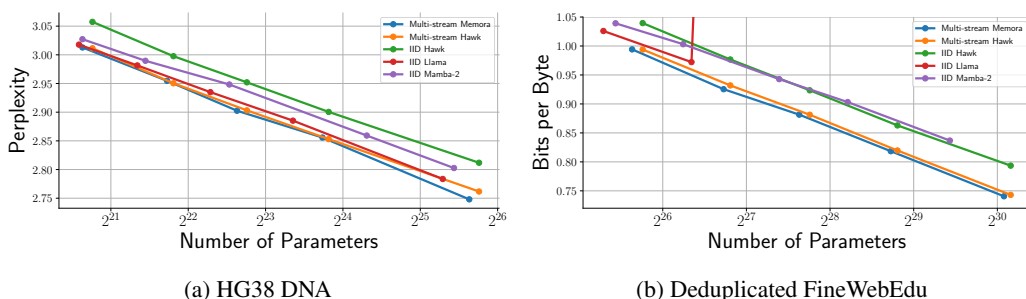

(a) HG38 DNA            (b) Deduplicated FineWebEdu

Figure 4: (A) Validation Perplexity scaling with number of parameters on the HG38 DNA dataset. (b) Validation Bits-Per-Byte scaling with the number of parameters on the de-duplicated FineWebEdu dataset. The segment length or the update stride is 1024.

forms both Hawk and Mamba2, demonstrating the strength of Transformer architectures. However, in the multi-stream setting, where a persistent state is maintained, Hawk surpasses Llama2 at most scales. Memora achieves the best overall performance, despite its simpler, recurrent-only design and lack of temporal convolutions, highlighting its memory capacity for capturing long-range structure.

## 5.4 BYTE-LEVEL LANGUAGE MODELING WITH FINEWEBEDU

Next, we study byte-level language modeling using a deduplicated version of the FineWebEdu dataset (Penedo et al. 2024), released as part of the SmolLM dataset (Allal et al. 2025). Byte-level modeling enables flexible, modality-agnostic sequence learning across domains like text, audio, and genomics by operating directly on raw bytes, removing the need for tokenization or domain-specific preprocessing. This approach can also improve generalization by avoiding biases introduced by methods like subword tokenization (Wang et al. 2024a). In Figure 4b, we present the performance of models trained for 150B bytes, and we show their model size scaling up to 1.2B parameters for multi-stream Memora, multi-stream Hawk, IID Llama2, IID Mamba2, and IID Hawk. We observe trends similar to the previous task, where multi-stream Hawk and Memora outperform IID Mamba2 in all scales, with Memora slightly outperforming Hawk. However, we notice that IID Llama2 starts to diverge when scaling up with byte-level data, indicating that standard Transformers struggle with raw bytes and may require additional components (e.g., Yu et al. 2023, Pagnoni et al. 2024).

## 5.5 LEARNING FROM ONE BYTE AT A TIME

Next, we study the challenging problem of learning language models using one sample (e.g., a character) at a time in an online fashion, the canonical learning mode of RTRL. Specifically, we consider a multi-stream setting in which the model receives one data point per time step from $B$ parallel streams and investigate the cases of $B = 1024$ and $B = 1$.

In Figure 5a, we evaluate language modeling performance using a single update stride with 1024 parallel streams. We compare Hawk and Memora, each tested under two modes: RTRL and 1-step BPTT. We find that both modes of Memora outperform their Hawk counterparts. Moreover, Memora performs better with RTRL than with 1-step BPTT, presumably because RTRL is able to assign credit over longer temporal dependencies, making it more effective for real-time learning.

Lastly, we investigate streaming sequence learning, where the model is updated at every time step with an update stride of 1 and using a single continuous stream of data. This streaming setting, commonly explored in prior work (e.g., Goyal et al. 2009, Elsayed et al. 2024), is particularly suited for on-device learning and real-time adaptation to non-stationary inputs, such as fine-tuning a pretrained model during deployment. Here, we tackle the challenging setting of streaming learning from scratch using byte-level inputs to demonstrate the viability of Memora.

Figure 5b compares the performance of Memora trained with RTRL against 1-step BPTT over 20M iterations. Both methods reduce the validation loss, but RTRL consistently outperforms 1-step BPTT. We note here that we train both systems for 20M bytes, where the dataset is about 2B bytes, which means both systems are severely undertrained. Nonetheless, to our knowledge, this is the first

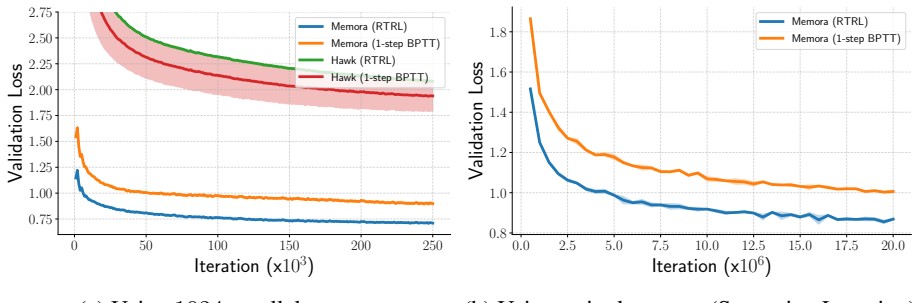

(a) Using 1024 parallel streams

(b) Using a single stream (Streaming Learning)

Figure 5: Learning from one byte at a time with a stride of $S=T=1$ using the TinyStories dataset, comparing Hawk and Memora under the multi-stream setting with RTRL against 1-step BPTT.

successful demonstration of deep sequence learning under strict streaming constraints. This result underscores the strong capacity of Memora to learn under stringent online learning constraints.

### 5.6 CONSIDERING GLRU ALTERNATIVES

Lastly, we assess the importance of our GLRU unit on a language modeling task using the TinyStories dataset with models of approximately 50M parameters. Since GLRU is a real-valued recurrent unit, we introduce a complex-valued variant by incorporating complex-valued gating in the recurrence, referred to as complex GLRU. We also compare against complex LRU (Orvieto et al. 2023) and its real-valued counterpart, real LRU, as well as MinGRU (Feng et al. 2024). We also evaluate restricted variants of complex GLRU and complex LRU, where the complex values are constrained as conjugate pairs (see Appendix D and G.5 for more details).

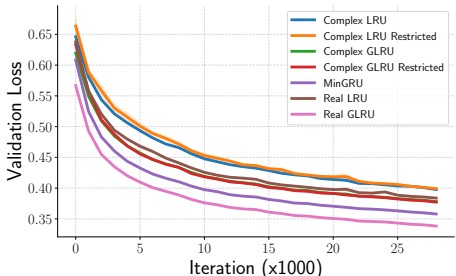

Figure 6: Performance of Memora with different recurrent units, including real-valued, complex-valued, gated, and non-gated.

Figure 6 presents validation performance across all baselines. Our model, real GLRU, consistently outperforms both gated (e.g., MinGRU) and non-gated (e.g., LRU) alternatives. We also find that constraining complex values to be conjugate pairs offers no performance benefit. Our findings align with observations by Gu and Dao (2024), where real-valued gated units surpass complex-valued ones on language tasks, further validating our results in this setting.

## 6 CONCLUSION, LIMITATIONS, AND FUTURE WORKS

This paper challenged the IID training paradigm for sequence learning and introduced multi-stream sequence modeling, a framework that preserves temporal continuity by presenting data in natural order and resetting only at meaningful boundaries (e.g., end of document). To support this paradigm, we proposed Memora, a lightweight recurrent architecture designed to maintain a persistent state across long sequences. Our results demonstrated that Memora under the multi-stream setting effectively models long-range dependencies and uniquely supports learning at extremely short update strides, including stride of 1. Within the sub-1.2B parameter regime, Memora consistently outperforms strong baselines, including Transformers, highlighting the potential of continuity-aware training and recurrence as a viable alternative to IID-based approaches.

While this work demonstrates the effectiveness of multi-stream sequence learning and the Memora architecture on long-range language modeling tasks, several limitations remain. Our experiments are limited to models with up to 1.2B parameters due to computational constraints typical in academic research settings, and it remains an open question whether the observed gains persist at larger scales. The scope of our evaluation is also restricted to byte-level language modeling; extending the paradigm to other modalities such as audio, video, or vision is an important direction for future work. Finally, we focus exclusively on next-byte prediction with cross-entropy loss, leaving the application of multi-stream training to settings requiring long-range credit assignment, such as reinforcement learning, for future research.

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

## A   RELATED WORKS

**Streaming learning.** Standard deep learning methods often assume access to the entire dataset; however, real-world applications require continuous data streams. There are a few supervised deep learning methods that work under the streaming learning setting (Hayes et al. 2019, Saran et al. 2023, Hayes and Kanan 2022), and additional efforts have adapted reinforcement learning (Elsayed et al. 2024, Vasan et al. 2024), language models (Goyal et al. 2009), and video predictors (Carreira et al. 2024, Qian et al. 2024, Han et al. 2025) to work under this setting. In our paper, we consider the multi-stream setting where there is more than one stream to process in parallel. Future work is needed to make Memora work with a single stream and a single update stride.

**Attention-free models.** To avoid the quadratic training cost and linear inference cost of attention on long histories, recent methods employ linear recurrence for fixed-size state memory, achieving linear training cost and constant inference cost. Mamba1 (Gu and Dao 2024), an approach that

combines a state-space approach (e.g., Gu et al. 2021) with temporal convolution (Bai et al. 2018), was the first method to be used in large-scale systems (Lieber et al. 2024), followed by Mamba2 (Dao and Gu 2024) and Hawk (De et al. 2024), which was based on the LRU recurrent unit (Orvieto et al. 2023). These architectures, however, still use IID chunking. In contrast, Memora is purely recurrent, preserving stream continuity to rival Transformers on long-sequence tasks.

**Byte-level data modeling.** Byte-level sequence modeling offers a flexible, domain-agnostic approach across text, audio, and genomics by operating directly on raw bytes, removing the need for preprocessing or tokenization. This can improve generalization and robustness to morphological variations like typos or character-level reasoning (Xue et al. 2022). However, it also introduces challenges due to longer, noisier, and less structured sequences. While recent works have improved byte-level modeling through architectural innovations (Wang et al. 2024a, Yu et al. 2023), many still rely on static chunking and overlook the temporal dynamics of streaming data. In contrast, our approach enhances byte-level modeling by processing streams in their natural order, improving memory and temporal reasoning capabilities critical for long, unsegmented byte sequences.

# B  THE T-BPTT ALGORITHM UNDER MULTI-STREAM SEQUENCE LEARNING

---

**Algorithm 1** Truncated BPTT with multi-stream sequence learning

---

1: **Require**: Number of Streams $B$, Learning update stride $T$
2: **Require**: Recurrent function $f_{\boldsymbol{\theta}}$ with parameters $\boldsymbol{\theta}$
3: **Require**: Output function $g_{\boldsymbol{\phi}}$ with parameters $\boldsymbol{\phi}$
4: **Require**: Data streams $\mathcal{D}_k, \forall k \in \{1, \ldots, B\}$, step size $\alpha$
5: **Initialize**: $\boldsymbol{\theta}, \boldsymbol{\phi}$, hidden state $\mathbf{h}_0 \leftarrow \mathbf{0}$
6: **for** $t_u = 1 \ldots \infty$ **do**                                                                    ▷ Update loop
7:   **for** $k = 1 \ldots B$ **do**                                    ▷ Go over streams (parallelizable due to independence)
8:     $\delta_{\boldsymbol{W}} \leftarrow \mathbf{0}, \forall \boldsymbol{W} \in \boldsymbol{\theta}, \quad \delta_{\boldsymbol{V}} \leftarrow \mathbf{0}, \forall \boldsymbol{V} \in \boldsymbol{\phi}, \quad \delta_{\boldsymbol{h}} \leftarrow \mathbf{0}$
9:     $\mathcal{B}_k \leftarrow \emptyset$                                         ▷ Buffer to store $(\mathbf{h}_{t-1}, \boldsymbol{x}_t, \mathbf{h}_t)$
10:    **for** $t = 1$ **upto** $T$ **do**                          ▷ Block item loop (parallelizable with parallel scan)
11:     $\boldsymbol{x}_t, \boldsymbol{y}_t, \texttt{reset} \leftarrow \mathcal{D}_k$
12:     **if** $\texttt{reset}$ **then**
13:      $\boldsymbol{h}_{t-1} \leftarrow \mathbf{0}$
14:     $\mathbf{h}_t \leftarrow f_{\boldsymbol{\theta}}(\mathbf{h}_{t-1}, \boldsymbol{x}_t)$                                            ▷ Forward pass
15:     $\hat{\boldsymbol{y}}_t \leftarrow g_{\boldsymbol{\phi}}(\mathbf{h}_t, \boldsymbol{x}_t)$
16:     Compute $\mathcal{L}_t = \mathcal{L}(\hat{\boldsymbol{y}}_t, \boldsymbol{y}_t)$
17:     $\delta_{\boldsymbol{V}} \leftarrow \delta_{\boldsymbol{V}} + \left(\frac{\partial \hat{\boldsymbol{y}}_t}{\partial \boldsymbol{V}}\right)^{\top} \frac{\partial \mathcal{L}_t}{\partial \hat{\boldsymbol{y}}_t}, \quad \boldsymbol{V} \in \boldsymbol{\phi}$
18:     $\delta_{\mathbf{h}} \leftarrow \delta_{\mathbf{h}} + \left(\frac{\partial \hat{\boldsymbol{y}}_t}{\partial \mathbf{h}_t}\right)^{\top} \frac{\partial \mathcal{L}_t}{\partial \hat{\boldsymbol{y}}_t}$
19:     Append $(\mathbf{h}_{t-1}, \boldsymbol{x}_t, \mathbf{h}_t)$ to $\mathcal{B}_k$
20:    **for** $i = t$ **downto** $1$ **do**                          ▷ Truncation loop (parallelizable with parallel scan)
21:     Retrieve $(\mathbf{h}_{i-1}, \boldsymbol{x}_i, \mathbf{h}_i)$ from $\mathcal{B}_k$
22:     $\delta_{\boldsymbol{W}} \leftarrow \delta_{\boldsymbol{W}} + \left(\frac{\partial \mathbf{h}_i}{\partial \boldsymbol{W}}\right)^{\top} \delta_{\mathbf{h}} \cdot, \quad \boldsymbol{W} \in \boldsymbol{\theta}$
23:     $\delta_{\mathbf{h}} \leftarrow \left(\frac{\partial \mathbf{h}_i}{\partial \mathbf{h}_{i-1}}\right)^{\top} \delta_{\mathbf{h}}$
24:   $\boldsymbol{W} \leftarrow \boldsymbol{W} - \frac{1}{BT} \alpha \, \delta_{\boldsymbol{W}}, \quad \boldsymbol{W} \in \boldsymbol{\theta}$                          ▷ One update after processing $B$ streams
25:   $\boldsymbol{V} \leftarrow \boldsymbol{V} - \frac{1}{BT} \alpha \, \delta_{\boldsymbol{V}}, \quad \boldsymbol{V} \in \boldsymbol{\phi}$

---

## C   THE RTRL ALGORITHM UNDER MULTI-STREAM SEQUENCE LEARNING

---

**Algorithm 2** RTRL with multi-stream sequence learning

---

1: **Require**: Number of streams $B$
2: **Require**: Recurrent function $f_{\boldsymbol{\theta}}$ with parameters $\boldsymbol{\theta}$
3: **Require**: Output function $g_{\boldsymbol{\phi}}$ with parameters $\boldsymbol{\phi}$
4: **Require**: Data streams $\mathcal{D}_k, \forall k \in \{1, \ldots, B\}$, step size $\alpha$
5: **Initialize**: $\boldsymbol{\theta}, \boldsymbol{\phi}$, hidden state $\mathbf{h}_0 \leftarrow \mathbf{0}$
6: **Initialize**: Sensitivity matrix $\boldsymbol{I}_0^{\boldsymbol{W}} \leftarrow \mathbf{0}, \forall \boldsymbol{W} \in \boldsymbol{\theta}$   $\qquad\qquad\qquad\qquad \triangleright \boldsymbol{I}_0^{\boldsymbol{W}} = \frac{\mathrm{d}\mathbf{h}_0}{\mathrm{d}\boldsymbol{W}}$
7: **for** $t = 1 \ldots \infty$ **do**
8:   $\delta_{\boldsymbol{V}} \leftarrow \mathbf{0}, \forall \boldsymbol{V} \in \boldsymbol{\phi}, \delta_{\boldsymbol{W}} \leftarrow \mathbf{0}, \forall \boldsymbol{V} \in \boldsymbol{\theta}$
9:   **for** $k = 1 \ldots B$ **do**   $\qquad\qquad\qquad\qquad\qquad\qquad\qquad\qquad\qquad\qquad \triangleright$ Go over streams
10:    $\boldsymbol{x}_t, \boldsymbol{y}_t, \texttt{reset} \leftarrow \mathcal{D}_{k,t}$
11:    **if** $\texttt{reset}$ **then**
12:     $\mathbf{h}_{t-1} \leftarrow \mathbf{0}$
13:    $\mathbf{h}_t \leftarrow f_{\theta}(\mathbf{h}_{t-1}, \mathbf{x}_t)$   $\qquad\qquad\qquad\qquad\qquad\qquad\qquad\qquad\qquad \triangleright$ Forward pass
14:    Update sensitivity: $\boldsymbol{I}_t^{\boldsymbol{W}} \leftarrow \frac{\partial \mathbf{h}_t}{\partial \boldsymbol{W}} + \frac{\partial \mathbf{h}_t}{\partial \mathbf{h}_{t-1}} \boldsymbol{I}_{t-1}^{\boldsymbol{W}}, \quad \forall \boldsymbol{W} \in \boldsymbol{\theta}$
15:    $\hat{\mathbf{y}}_t \leftarrow g_{\phi}(\mathbf{h}_t, \mathbf{x}_t)$
16:    Compute $\mathcal{L}_t = \mathcal{L}(\hat{\boldsymbol{y}}_t, \boldsymbol{y}_t)$
17:    $\delta_{\boldsymbol{V}} \leftarrow \delta_{\boldsymbol{V}} + \frac{1}{B} \left( \frac{\partial \hat{\mathbf{y}}_t}{\partial \boldsymbol{V}} \right)^{\top} \frac{\partial \mathcal{L}_t}{\partial \hat{\mathbf{y}}_t}, \quad \forall \boldsymbol{V} \in \boldsymbol{\phi}$   $\qquad\qquad\qquad \triangleright$ Gradient for $\boldsymbol{\phi}$
18:    $\delta_{\boldsymbol{W}} \leftarrow \delta_{\boldsymbol{W}} + \frac{1}{B} \left( \boldsymbol{I}_t^{\boldsymbol{W}} \right)^{\top} \left( \frac{\partial \hat{\mathbf{y}}_t}{\partial \mathbf{h}_t} \right)^{\top} \frac{\partial \mathcal{L}_t}{\partial \hat{\mathbf{y}}_t}, \quad \forall \boldsymbol{W} \in \boldsymbol{\theta}$   $\qquad \triangleright$ Gradient for $\boldsymbol{\theta}$
19:   $\boldsymbol{W} \leftarrow \boldsymbol{W} - \alpha\, \delta_{\boldsymbol{W}}, \quad \forall \boldsymbol{W} \in \boldsymbol{\theta}$   $\qquad\qquad\qquad \triangleright$ Parameter update for $\boldsymbol{\theta}$
20:   $\boldsymbol{V} \leftarrow \boldsymbol{V} - \alpha\, \delta_{\boldsymbol{V}}, \quad \forall \boldsymbol{V} \in \boldsymbol{\phi}$   $\qquad\qquad\qquad\quad \triangleright$ Parameter update for $\boldsymbol{\phi}$

---

## D   COMPLEX-VALUED GATED LINEAR RECURRENT UNIT

Here, we describe the complex-valued GLRU unit. We replace our real-valued recurrent gating and input gating with their complex counterparts.

$$\boldsymbol{h}_t = \boldsymbol{\lambda}(\boldsymbol{x}_t) \circ \boldsymbol{h}_{t-1} + \boldsymbol{\gamma}_t \circ \boldsymbol{g}(\boldsymbol{x}_t) \circ (\boldsymbol{B}\boldsymbol{x}_t)$$
$$\boldsymbol{y}_t = \Re[\boldsymbol{C}\boldsymbol{h}_t],$$

where $\boldsymbol{\lambda}(\boldsymbol{x}_t) \doteq \boldsymbol{r}(\boldsymbol{x}_t) \circ e^{i\boldsymbol{\theta}_t}, \boldsymbol{\gamma}_t = \sqrt{1 - |\boldsymbol{\lambda}_t|}, \boldsymbol{g}(\boldsymbol{x}_t) = \boldsymbol{G}\boldsymbol{x}_t$, and $\boldsymbol{r}(\boldsymbol{x}_t) = e^{-c e^{\boldsymbol{\nu}} \circ \sigma(\boldsymbol{R}\boldsymbol{x}_t)}$. The vector $\boldsymbol{\theta}_t \in \mathbb{R}^n$ contains the phase information of the complex-valued system. We need to learn complex-valued matrices, $\boldsymbol{B} \in \mathbb{C}^{n \times d}, \boldsymbol{G} \in \mathbb{C}^{n \times d}$, and $\boldsymbol{C} \in \mathbb{C}^{m \times n}$. Note that $\boldsymbol{R} \in \mathbb{R}^{n \times n}$ and $\boldsymbol{\nu} \in \mathbb{R}^n$ are still real-valued. We have to use $\boldsymbol{y}_t = \Re[\boldsymbol{C}\boldsymbol{h}_t]$ instead of $\boldsymbol{y}_t = \Re[\boldsymbol{h}_t]$ to have no gradient bias. We refer the reader to Elelimy et al. (2024) and Orvieto et al. (2023) for more discussion about gradient bias. We implement complex-valued GLRU using the cosine representation with real-valued systems, and we refer the reader to Appendix G.4 for the details on how to convert the system from exponential representation form to cosine representation.

## E   RTRL MODE OF GLRU AND RG-LRU RECURRENT UNITS

Here, we derive the RTRL sensitivity update equations for GLRU and RG-LRU units. We show that in both, their sensitivity tensors are diagonal and can be stored and computed efficiently.

### E.1   GLRU RTRL SENSITIVITY EQUATIONS

The recurrence equation of GLRU is given by

$$\boldsymbol{h}_t = \boldsymbol{r}(\boldsymbol{x}_t) \circ \boldsymbol{h}_{t-1} + \boldsymbol{\gamma}_t \circ \boldsymbol{g}(\boldsymbol{x}_t) \circ (\boldsymbol{B}\boldsymbol{x}_t)$$
$$\boldsymbol{y}_t = \boldsymbol{h}_t$$

where $\gamma_t = \sqrt{1 - r_t^2}$, $g(x_t) = Gx_t$, and $r(x_t) = e^{-ce^{\nu} \circ \sigma(Rx_t)}$. Now, we can derive the sensitivity update equation for the vector $\nu$ using index notations as follows:

$$
S_{t,i,j}^{\nu} = \frac{\partial h_{t,i}}{\partial \nu_j} = \frac{\partial}{\partial \nu_j} \left( r_i h_{t-1,i} + \gamma_i g_i \left( \sum_m B_{i,m} x_{t,m} \right) \right)
$$

$$
= \frac{\partial r_i}{\partial \nu_j} h_{t-1,i} + r_i S_{t-1,i,j}^{\nu} + \frac{\partial \gamma_i}{\partial \nu_j} g_i \left( \sum_m B_{i,m} x_{t,m} \right)
$$

$$
= -c\delta_{i,j} e^{\nu_i} \sigma \left( \sum_k R_{i,m} x_{t,m} \right) r_i h_{t-1,i} + r_i S_{t-1,i,j}^{\nu}
$$

$$
+ c\delta_{i,j} \frac{r_i^2}{\gamma_i} e^{\nu_i} \sigma \left( \sum_k R_{i,m} x_{t,m} \right) g_i \left( \sum_m B_{i,m} x_{t,m} \right)
$$

Note how the structure coming from $\delta_{i,j}$ forces all off-diagonal elements where $i \neq j$ to be zero. Hence, we can store the sensitivity elements in a vector instead of a matrix.

We can also write the recursive relationship using the reduced sensitivity for $\nu$ as follows:

$$
\mathbf{S}_t^{\nu} = \frac{\partial}{\partial \nu} \left( r(x_t) \circ h_{t-1} + \gamma \circ g(x_t) \circ (Bx_t) \right)
$$

$$
= r(x_t) \circ \mathbf{S}_{t-1}^{\nu} - ce^{\nu} \circ \sigma(Rx_t) \circ r(x_t) \circ h_{t-1} + c\frac{r(x_t)^2}{\gamma} \circ e^{\nu} \circ \sigma(Rx_t) \circ g(x_t) \circ (Bx_t)
$$

where $[\mathbf{S}_t^{\nu}]_i \doteq [S_{t,i,j}^{\nu}]_{i,j=1}$.

Next, we derive the sensitivity equation for the matrix $B$ as follows:

$$
S_{t,i,j,k}^{G} = \frac{\partial h_{t,i}}{\partial G_{j,k}} = \frac{\partial}{\partial G_{j,k}} \left( r_i h_{t-1,i} + \gamma_i b_i \sum_m G_{i,m} x_{t,m} \right)
$$

$$
= r_i S_{t-1,i,j,k}^{G} \delta_{i,j} + \gamma_i b_i \delta_{i,j} x_{t,k},
$$

where we use $b_i = \sum_k B_{i,k} x_{t,k}$. Note how the structure coming from $\delta_{i,j}$ forces all off-diagonal elements where $i \neq j$ to be zero. Hence, we can store the sensitivity elements in a matrix instead of a 3-tensor. We can also write the recursive relationship using the reduced sensitivity objects as follows:

$$
\mathbf{S}_t^{G} = \frac{\partial}{\partial G} \left( r(x_t) \circ h_{t-1} + \gamma \circ g(x_t) \circ (Gx_t) \right)
$$

$$
= \mathrm{Diag}(r(x_t)) \circ \mathbf{S}_{t-1}^{G} + (\gamma \circ Bx_t) x_t^{\top},
$$

where $[\mathbf{S}_t^{G}]_{i,j} \doteq [S_{t,i,j,k}^{G}]_{i,j,k=1}$.

Next, we derive the sensitivity equation for the matrix $B$ as follows:

$$
S_{t,i,j,k}^{B} = \frac{\partial h_{t,i}}{\partial B_{j,k}} = \frac{\partial}{\partial B_{j,k}} \left( r_i h_{t-1,i} + \gamma_i g_i \sum_m B_{i,m} x_{t,m} \right)
$$

$$
= r_i S_{t-1,i,j,k}^{B} \delta_{i,j} + \gamma_i g_i \sum_m \delta_{i,j} \delta_{k,m} x_{t,m}
$$

$$
= r_i S_{t-1,i,j,k}^{B} \delta_{i,j} + \gamma_i g_i \delta_{i,j} x_{t,k}.
$$

Note how the structure coming from $\delta_{i,j}$ forces all off-diagonal elements where $i \neq j$ to be zero. Hence, we can store the sensitivity elements in a matrix instead of a 3-tensor. We can also write the recursive relationship using the reduced sensitivity objects as follows:

$$
\mathbf{S}_t^{B} = \frac{\partial}{\partial B} \left( r(x_t) \circ h_{t-1} + \gamma \circ g(x_t) \circ (Bx_t) \right)
$$

$$
= \mathrm{Diag}(r(x_t)) \circ \mathbf{S}_{t-1}^{B} + (\gamma \circ g(x_t)) x_t^{\top},
$$

where $\left[\mathbf{S}_t^{\boldsymbol{B}}\right]_{i,j} \doteq \left[S_{t,i,j,k}^{\boldsymbol{B}}\right]_{i,j,k=1}$.

Finally, we derive the sensitivity equation for the matrix $\boldsymbol{R}$ as follows:

$$S_{t,i,j,k}^{\boldsymbol{R}} = \frac{\partial h_{t,i}}{\partial R_{j,k}} = \frac{\partial}{\partial R_{j,k}} \left( r_i h_{t-1,i} + \gamma_i g_i \sum_m B_{i,m} x_{t,m} \right)$$

$$= \delta_{i,j} r_i S_{t-1,i,j}^{\boldsymbol{R}} + \frac{\partial r_i}{\partial R_{j,k}} h_{t-1,i}$$

$$= \delta_{i,j} r_i S_{t-1,i,j}^{\boldsymbol{R}} - c\delta_{i,j} r_i e^{\nu_i} \sigma \left( \sum_k R_{i,m} x_{t,m} \right) \left( 1 - \sigma' \left( \sum_k R_{i,m} x_{t,m} \right) \right) h_{t-1,i} x_{t,k}$$

$$- c\delta_{i,j} r_i e^{\nu_i} \sigma \left( \sum_k R_{i,m} x_{t,m} \right) \left( 1 - \sigma' \left( \sum_k R_{i,m} x_{t,m} \right) \right) \frac{r_i}{\gamma_i} g_i \left( \sum_m B_{i,m} x_{t,m} \right) x_{t,k}$$

$$= \delta_{i,j} r_i S_{t-1,i,j}^{\boldsymbol{R}} - \delta_{i,j} d_i \left( h_{t-1,i} - \frac{r_i}{\gamma_i} g_i \sum_m B_{i,m} x_{t,m} \right) x_{t,k},$$

where $d_i = c r_i e^{\nu_i} \sigma \left( \sum_k R_{i,m} x_{t,m} \right) \left( 1 - \sigma' \left( \sum_k R_{i,m} x_{t,m} \right) \right)$. Note how the structure coming from $\delta_{i,j}$ forces all off-diagonal elements where $i \neq j$ to be zero. Hence, we can store the sensitivity elements in a matrix instead of a 3-tensor. We can also write the recursive relationship using the reduced sensitivity objects as follows:

$$\mathbf{S}_t^{\boldsymbol{R}} = \frac{\partial}{\partial \boldsymbol{R}} \left( \boldsymbol{r}(\boldsymbol{x}_t) \circ \boldsymbol{h}_{t-1} + \boldsymbol{\gamma} \circ \boldsymbol{g}(\boldsymbol{x}_t) \circ (\boldsymbol{B}\boldsymbol{x}_t) \right)$$

$$= \mathrm{Diag}(\boldsymbol{r}(\boldsymbol{x}_t)) \boldsymbol{S}_r^{\boldsymbol{R}} + \left( \boldsymbol{d} \circ \left( \boldsymbol{h}_{t-1} - \frac{\boldsymbol{r}(\boldsymbol{x}_t)}{\boldsymbol{\gamma}} \circ \boldsymbol{g}(\boldsymbol{x}_t) \circ (\boldsymbol{B}\boldsymbol{x}_t) \right) \right) \boldsymbol{x}_t^{\top}.$$

where $\boldsymbol{d} = c\boldsymbol{r}(\boldsymbol{x}_t) \circ e^{\boldsymbol{\nu}} \circ \sigma(\boldsymbol{R}\boldsymbol{x}_t) \circ (1 - \sigma'(\boldsymbol{R}\boldsymbol{x}_t))$ and $\left[\mathbf{S}_t^{\boldsymbol{R}}\right]_{i,j} \doteq \left[S_{t,i,j,k}^{\boldsymbol{R}}\right]_{i,j,k=1}$.

### E.2 RG-LRU RTRL SENSITIVITY EQUATIONS

The RG-LRU unit was introduced by De et al. (2024) and is typically used with T-BPTT. Here, we derive its RTRL mode. We start by writing the RG-LRU unit in the same notation we use in this paper. The RG-LRU unit is given by

$$\boldsymbol{h}_t = \boldsymbol{r}(\boldsymbol{x}_t) \circ \boldsymbol{h}_{t-1} + \boldsymbol{\gamma}_t \circ \boldsymbol{g}(\boldsymbol{x}_t) \circ \boldsymbol{x}_t$$
$$\boldsymbol{y}_t = \boldsymbol{h}_t$$

where $\boldsymbol{\gamma}_t = \sqrt{1 - \boldsymbol{r}_t^2}, \boldsymbol{g}(\boldsymbol{x}_t) = \sigma(\boldsymbol{G}\boldsymbol{x}_t), \boldsymbol{r}(\boldsymbol{x}_t) = e^{-c \log(1+e^{\boldsymbol{\nu}}) \circ \sigma(\boldsymbol{R}\boldsymbol{x}_t)}$

The sensitivity update equation for the vector $\boldsymbol{\nu}$ are given by:

$$S_{t,i,j}^{\boldsymbol{\nu}} = \frac{\partial h_{t,i}}{\partial \nu_j} = \frac{\partial}{\partial \nu_j} \left( r_i h_{t-1,i} + \gamma_i g_i \left( \sum_m B_{i,m} x_{t,m} \right) \right)$$

$$= \frac{\partial r_i}{\partial \nu_j} h_{t-1,i} + r_i S_{t-1,i,j}^{\boldsymbol{\nu}} + \frac{\partial \gamma_i}{\partial \nu_j} g_i \left( \sum_m B_{i,m} x_{t,m} \right)$$

$$= -c\delta_{i,j} \sigma(\nu_i) \sigma \left( \sum_k R_{i,m} x_{t,m} \right) r_i h_{t-1,i} + r_i S_{t-1,i,j}^{\boldsymbol{\nu}}$$

$$+ c\delta_{i,j} \frac{r_i^2}{\gamma_i} \sigma(\nu_i) \sigma \left( \sum_k R_{i,m} x_{t,m} \right) g_i x_{t,i}$$

Note how the structure coming from $\delta_{i,j}$ forces all off-diagonal elements where $i \neq j$ to be zero. Hence, we can store the sensitivity elements in a vector instead of a matrix.

We can also write the recursive relationship using the reduced sensitivity for $\boldsymbol{\nu}$ as follows:

$$\mathbf{S}_t^{\boldsymbol{\nu}} = \frac{\partial}{\partial \boldsymbol{\nu}} \left( \boldsymbol{r}(\boldsymbol{x}_t) \circ \boldsymbol{h}_{t-1} + \boldsymbol{\gamma} \circ \boldsymbol{g}(\boldsymbol{x}_t) \circ \boldsymbol{x}_t \right)$$

$$= \boldsymbol{r}(\boldsymbol{x}_t) \circ \mathbf{S}_{t-1}^{\boldsymbol{\nu}} - c\sigma(\boldsymbol{\nu}) \circ \sigma(\boldsymbol{R}\boldsymbol{x}_t) \circ \boldsymbol{r}(\boldsymbol{x}_t) \circ \boldsymbol{h}_{t-1} + c\frac{\boldsymbol{r}(\boldsymbol{x}_t)^2}{\boldsymbol{\gamma}} \circ \sigma(\boldsymbol{\nu}) \circ \sigma(\boldsymbol{R}\boldsymbol{x}_t) \circ \boldsymbol{g}(\boldsymbol{x}_t) \circ \boldsymbol{x}_t,$$

where $[\mathbf{S}_t^{\boldsymbol{\nu}}]_i \doteq \left[ S_{t,i,j}^{\boldsymbol{\nu}} \right]_{i,j=1}$.

Next, we derive the sensitivity equation for the matrix $\boldsymbol{G}$ as follows:

$$S_{t,i,j,k}^{\boldsymbol{G}} = \frac{\partial h_{t,i}}{\partial G_{j,k}} = \frac{\partial}{\partial G_{j,k}} \left( r_i h_{t-1,i} + \gamma_i x_i \sum_m G_{i,m} x_{t,m} \right)$$

$$= r_i S_{t-1,i,j,k}^{\boldsymbol{G}} \delta_{i,j} + \gamma_i x_i \delta_{i,j} x_{t,k},$$

Note how the structure coming from $\delta_{i,j}$ forces all off-diagonal elements where $i \neq j$ to be zero. Hence, we can store the sensitivity elements in a matrix instead of a 3-tensor. We can also write the recursive relationship using the reduced sensitivity objects as follows:

$$\mathbf{S}_t^{\boldsymbol{G}} = \frac{\partial}{\partial \boldsymbol{G}} \left( \boldsymbol{r}(\boldsymbol{x}_t) \circ \boldsymbol{h}_{t-1} + \boldsymbol{\gamma} \circ \boldsymbol{g}(\boldsymbol{x}_t) \circ \boldsymbol{x}_t \right)$$

$$= \mathrm{Diag}(\boldsymbol{r}(\boldsymbol{x}_t)) \circ \mathbf{S}_{t-1}^{\boldsymbol{G}} + (\boldsymbol{\gamma} \circ \boldsymbol{x}_t) \boldsymbol{x}_t^{\top}.$$

where $\left[ \mathbf{S}_t^{\boldsymbol{G}} \right]_{i,j} \doteq \left[ S_{t,i,j,k}^{\boldsymbol{G}} \right]_{i,j,k=1}$.

Finally, we derive the sensitivity equation for the matrix $\boldsymbol{R}$ as follows:

$$S_{t,i,j,k}^{\boldsymbol{R}} = \frac{\partial h_{t,i}}{\partial R_{j,k}} = \frac{\partial}{\partial R_{j,k}} \left( r_i h_{t-1,i} + \gamma_i g_i x_{t,i} \right)$$

$$= \delta_{i,j} r_i S_{t-1,i,j}^{\boldsymbol{R}} + \frac{\partial r_i}{\partial R_{j,k}} h_{t-1,i}$$

$$= \delta_{i,j} r_i S_{t-1,i,j}^{\boldsymbol{R}} - c\delta_{i,j} r_i \log(1 + e^{\nu_i}) \sigma\left( \sum_k R_{i,m} x_{t,m} \right) \left( 1 - \sigma'\left( \sum_k R_{i,m} x_{t,m} \right) \right) h_{t-1,i} x_{t,k}$$

$$- c\delta_{i,j} r_i \log(1 + e^{\nu_i}) \sigma\left( \sum_k R_{i,m} x_{t,m} \right) \left( 1 - \sigma'\left( \sum_k R_{i,m} x_{t,m} \right) \right) \frac{r_i}{\gamma_i} g_i x_{t,i} x_{t,k}$$

$$= \delta_{i,j} r_i S_{t-1,i,j}^{\boldsymbol{R}} - \delta_{i,j} d_i \left( h_{t-1,i} - \frac{r_i}{\gamma_i} g_i x_{t,i} \right) x_{t,k},$$

where $d_i = cr_i \log(1 + e^{\nu_i}) \sigma\left( \sum_k R_{i,m} x_{t,m} \right) \left( 1 - \sigma'\left( \sum_k R_{i,m} x_{t,m} \right) \right)$. Note how the structure coming from $\delta_{i,j}$ forces all off-diagonal elements where $i \neq j$ to be zero. Hence, we can store the sensitivity elements in a matrix instead of a 3-tensor. We can also write the recursive relationship using the reduced sensitivity objects as follows:

$$\mathbf{S}_t^{\boldsymbol{R}} = \frac{\partial}{\partial \boldsymbol{R}} \left( \boldsymbol{r}(\boldsymbol{x}_t) \circ \boldsymbol{h}_{t-1} + \boldsymbol{\gamma} \circ \boldsymbol{g}(\boldsymbol{x}_t) \circ (\boldsymbol{B}\boldsymbol{x}_t) \right)$$

$$= \mathrm{Diag}(\boldsymbol{r}(\boldsymbol{x}_t)) \boldsymbol{S}_r^{\boldsymbol{R}} + \left( \boldsymbol{d} \circ \left( \boldsymbol{h}_{t-1} - \frac{\boldsymbol{r}(\boldsymbol{x}_t)}{\boldsymbol{\gamma}} \circ \boldsymbol{g}(\boldsymbol{x}_t) \circ \boldsymbol{x}_t \right) \right) \boldsymbol{x}_t^{\top}.$$

where $\boldsymbol{d} = c\boldsymbol{r}(\boldsymbol{x}_t) \circ \log(1 + e^{\boldsymbol{\nu}}) \circ \sigma(\boldsymbol{R}\boldsymbol{x}_t) \circ (1 - \sigma'(\boldsymbol{R}\boldsymbol{x}_t))$ and $\left[ \mathbf{S}_t^{\boldsymbol{R}} \right]_{i,j} \doteq \left[ S_{t,i,j,k}^{\boldsymbol{R}} \right]_{i,j,k=1}$.

# F EXPERIMENTAL DETAILS

We use Python and Pytorch (Paszke et al. 2019) to implement our algorithms using automatic differentiation to backpropagate gradients with T-BPTT and RTRL. Additionally, we used the parallel scan implementation by Kyrylov (2024).

We used LeCun initialization (LeCun et al. 2012) to initialize all weights except for the weights used for contracting the input (see Figure 2), which we initialize it $W_{i,j} \sim \mathcal{N}(0, 1/\sqrt{2 \times E \times N})$, where $N$ is the number of layers in the model. Additionally, we use the ring initialization (Orvieto et al. 2023) in both RG-LRU and GLRU, using $r_{\min} = 0.9$, $r_{\max} = 0.999$, which is given as: $\boldsymbol{\nu}_{\text{init}} \leftarrow \log(-0.5\log(\boldsymbol{u}(r_{\max}^2 - r_{\min}^2) + r_{\min}^2)$, where $u_i \sim U[0,1], \forall i$. In our experiments, the truncation length in recurrent-based models is always equal to the segment length.

In Figure 3, we list the common training configurations we used in all experiments. We then describe the specific details for each experiment in the next sections.

| Configuration | Value |
|---|---|
| Optimizer | AdamW |
| Optimizer parameters | $\beta_1 = 0.9, \beta_2 = 0.95$ |
| Weight decay | 0.1 |
| Bias | No |
| Dropout | No |
| Gradient Clipping | 1.0 |
| Floating-point precision | Bfloat16 |
| GPU used | NVIDIA L40/H100 |
| Automatic mixed precision | Yes |
| Embedding Weight Tying (Press and Wolf 2016) | Yes |

Table 3: The common training configuration shared in all experiments and baselines.

## F.1 SELECTIVE COPYING

We trained all models for $50,000$ iterations using a batch size of $64$ and a constant learning rate of $3 \times 10^{-4}$, evaluating performance on $5,000$ randomly generated examples. Each model consists of two layers. The Memora variants—GLRU, MinGRU, and LRU—share the same configuration: a model dimension of 64, state dimension of 256, and a gated MLP with an expansion ratio of 3. Hawk uses a similar setup to Memora, with the addition of a convolution kernel size of 4. Llama2, by contrast, is configured with a larger model dimension of 192, three attention heads, and an MLP expansion ratio of 4. Mamba2 also adopts a model dimension of 192 but differs with a state dimension of 128, a head dimension of 64, an MLP expansion ratio of 2, and a convolution kernel size of 4. We report the average of 5 independent runs and report the standard error.

## F.2 BYTE-LEVEL LANGUAGE MODELING WITH TINYSTORIES

We use an effective batch size of $BT = 131,072$ and vary the sequence length in the IID setting or the update stride in the multi-stream setting with values $T \in \{2, 8, 32, 128, 512, 1024\}$. All models are trained for 2 epochs using a constant learning rate of $3 \times 10^{-4}$, and are configured to have approximately 60 million parameters. We used the UTF-8 character encoding that represents each character with one to four bytes. The Mamba2 model uses a model dimension of 768, a state dimension of 128, a head dimension of 128, 16 layers, a convolutional kernel size of 4, and an MLP expansion ratio of 2, totaling 60,294,464 parameters. The Hawk model is configured with a model dimension of 512, a state dimension of 768, 14 layers, a gated MLP expansion ratio of 3, and a convolutional kernel size of 4, totaling $66,259,968$ parameters. The Memora model uses a model dimension of 512, a state dimension of 768, 14 layers, and an MLP expansion ratio of 3, totaling 60,711,424 parameters. Finally, the Llama2 baseline has a model dimension of 512, 8 attention heads, 18 layers, and an MLP expansion ratio of 4, totaling 61,491,712 parameters. We report the average of 3 runs (the error bars are very small, so we do not display them to reduce clutter and enhance visibility).

## F.3 DNA MODELING

We use an effective batch size of $BT = 524,288$ with a batch size of $B = 512$ and trained for 4 epochs. We used a learning rate warm-up for $10\%$ of the total iterations, followed by cosine

annealing with a minimum of $10^{-5}$ and a maximum of $10^{-3}$. We compare Memora, Mamba2, Hawk, and Llama2, each of which use 5 different model sizes in each method (sizes with prefixes S0, S1, S2, S3, S4). We list the model configurations in Table 4.

In Mamba2 and Hawk, the temporal convolutional kernel size is 4. In Hawk and Memora, we use MLP expansion factor of 3 in the gated MLP blocks (by setting $E = 3 \times M$ in Figure 2). In Mamba2, the expansion factor is set to 2. In Llama2, the MlP expansion factor is set to 4. We report the average of 3 runs (the error bars are too small to be visible in the figure).

### F.4 BYTE-LEVEL LANGUAGE MODELING WITH FINEWEBEDU

We use an effective batch size of $BT = 524,288$ with a batch size of $B = 512$ and trained for $286,500$ iterations. We used a learning rate warm-up for $1\%$ of the total iterations, followed by cosine annealing with a minimum of $3 \times 10^{-5}$ and a maximum of $3 \times 10^{-4}$. We compare Memora, Mamba2, Hawk, and Llama2, each of which use 5 different model sizes in each method (sizes with prefixes S4, S5, S6, S7, S8). We list the model configurations in Table 4. We use the same other model parameters mentioned in the previous section.

### F.5 LEARNING FROM ONE SAMPLE AT A TIME WITH RTRL AND 1-STEP BPTT

For both Memora and Hawk, we use a model dimension of $384$, a state dimension of $512$, six layers, and a gated MLP expansion factor of 3. Training is conducted over 1024 parallel data streams with a single update stride for two epochs. We employ a constant learning rate and tune each method by selecting the best learning rate from the set $\{3 \times 10^{-4}, 3 \times 10^{-5}, 3 \times 10^{-6}, 3 \times 10^{-7}, 3 \times 10^{-8}\}$. The optimal learning rate was found to be $3 \times 10^{-5}$ for Memora (both RTRL and 1-BPTT variants) and $3 \times 10^{-7}$ for Hawk (for both RTRL and 1-BPTT). We report the average of 3 runs.

### F.6 MEMORA WITH STREAMING LEARNING

We use an effective batch size of $BT = T = B = 1$ and train for 20M iterations with a constant learning rate of $3 \times 10^{-5}$ for RTRL and $3 \times 10^{-7}$ for 1-step BPTT which was selected based on searching in $\{3 \times 10^{-4}, 3 \times 10^{-5}, 3 \times 10^{-6}, 3 \times 10^{-7}\}$. We used a model dimension of $512$, a state dimension of $768$, a gated MLP expansion factor of 3, and 6 Memora layers. We used RMSProp with $\beta_2 = 0.9999$ with no weight decay nor gradient clipping. We report the average of 3 runs.

### F.7 MEMORA WITH GLRU ALTERNATIVES

We use an effective batch size of $BT = 131,072$, with a batch size of $B = 256$. We train for 2 epochs using a constant learning rate of $3 \times 10^{-4}$. We use a model dimension of $512$, a state dimension of $768$, 12 layers, and an MLP expansion factor of 3. We report the average of 3 runs.

## G   PRIMER ON LINEAR RECURRENT UNITS

The learner usually observes the environment partially; thus, it is required to construct its *learner state*, some internal representation of what the state of the environment might be. We denote the state construction function $f : \mathbb{R}^n \times \mathbb{R}^d \to \mathbb{R}^n$ given by $\boldsymbol{h}_t = f(\boldsymbol{h}_{t-1}, \boldsymbol{x}_t)$, where $\boldsymbol{h}_t \in \mathbb{R}^n$ and $\boldsymbol{x}_t \in \mathbb{R}^d$ are the learner state and observation at time step $t$. The learner state is considered the learner's best ability to construct a compact history of the past. The output construction function $g : \mathbb{R}^d \times \mathbb{R}^n \to \mathbb{R}^m$ maps the learner state into some usable output $\boldsymbol{y}_t \in \mathbb{R}^m$ for prediction and is given by $\boldsymbol{y}_t = g(\boldsymbol{h}_t, \boldsymbol{x}_t)$. The evolution of the system is fully described using the following:

$$\boldsymbol{h}_t = f(\boldsymbol{h}_{t-1}, \boldsymbol{x}_t),$$
$$\boldsymbol{y}_t = g(\boldsymbol{h}_t, \boldsymbol{x}_t).$$

In the simple case of a linear system, the equations can be formulated as

$$\boldsymbol{h}_t = \boldsymbol{A}\boldsymbol{h}_{t-1} + \boldsymbol{B}\boldsymbol{x}_t$$
$$\boldsymbol{y}_t = \boldsymbol{C}\boldsymbol{h}_t + \boldsymbol{D}\boldsymbol{x}_t,$$

| Model | Params | Model Dim | State Dim | Heads | Head Dim | Layers (N) |
|---|---|---|---|---|---|---|
| **Memora-S8** | 1,151,980,544 | 2048 | 2560 | – | – | 18 |
| **Memora-S7** | 444,002,304 | 1536 | 2048 | – | – | 12 |
| **Memora-S6** | 207,923,200 | 1024 | 1536 | – | – | 12 |
| **Memora-S5** | 111,114,240 | 768 | 1024 | – | – | 12 |
| **Memora-S4** | 52,057,088 | 512 | 768 | – | – | 12 |
| **Memora-S3** | 13,966,848 | 384 | 512 | – | – | 6 |
| **Memora-S2** | 6,495,232 | 256 | 384 | – | – | 6 |
| **Memora-S1** | 3,470,400 | 192 | 256 | – | – | 6 |
| **Memora-S0** | 1,625,600 | 128 | 192 | – | – | 6 |
| **Hawk-S8** | 1,199,352,832 | 2048 | 2560 | – | – | 18 |
| **Hawk-S7** | 469,267,968 | 1536 | 2048 | – | – | 12 |
| **Hawk-S6** | 226,872,320 | 1024 | 1536 | – | – | 12 |
| **Hawk-S5** | 117,455,616 | 768 | 1024 | – | – | 12 |
| **Hawk-S4** | 56,813,056 | 512 | 768 | – | – | 12 |
| **Hawk-S3** | 14,765,952 | 384 | 512 | – | – | 6 |
| **Hawk-S2** | 7,094,528 | 256 | 384 | – | – | 6 |
| **Hawk-S1** | 3,673,344 | 192 | 256 | – | – | 6 |
| **Hawk-S0** | 1,777,792 | 128 | 192 | – | – | 6 |
| **Llama2-S8** | 906,493,952 | 2048 | – | 16 | – | 18 |
| **Llama2-S7** | 340,170,240 | 1536 | – | 16 | – | 12 |
| **Llama2-S6** | 154,428,416 | 1024 | – | 16 | – | 12 |
| **Llama2-S5** | 85,150,464 | 768 | – | 12 | – | 12 |
| **Llama2-S4** | 41,038,336 | 512 | – | 8 | – | 12 |
| **Llama2-S3** | 10,720,128 | 384 | – | 6 | – | 6 |
| **Llama2-S2** | 5,116,928 | 256 | – | 4 | – | 6 |
| **Llama2-S1** | 2,658,048 | 192 | – | 4 | – | 6 |
| **Llama2-S0** | 1,575,424 | 128 | – | 4 | – | 6 |
| **Mamba2-S8** | 724,556,000 | 2560 | 128 | – | 64 | 18 |
| **Mamba2-S7** | 310,717,696 | 2048 | 128 | – | 64 | 12 |
| **Mamba2-S6** | 176,124,096 | 1536 | 128 | – | 64 | 12 |
| **Mamba2-S5** | 79,475,840 | 1024 | 128 | – | 64 | 12 |
| **Mamba2-S4** | 45,381,216 | 768 | 128 | – | 64 | 12 |
| **Mamba2-S3** | 20,772,928 | 512 | 128 | – | 64 | 12 |
| **Mamba2-S2** | 6,062,424 | 384 | 128 | – | 64 | 6 |
| **Mamba2-S1** | 2,858,384 | 256 | 128 | – | 64 | 6 |
| **Mamba2-S0** | 1,625,600 | 192 | 128 | – | 64 | 6 |

Table 4: Model configurations of Memora, Hawk, Llama2, and Mamba2 models used in the DNA modeling and de-duplicated FineWebEdu experiments

where $A \in \mathbb{R}^{n \times n}$, $B \in \mathbb{R}^{n \times d}$, $C \in \mathbb{R}^{n \times m}$, and $D \in \mathbb{R}^{d \times m}$.

We can write the square matrix $A$ using its eigenvalue decomposition as $A = P^{-1}\Lambda P$, where $P \in \mathbb{C}^{n \times n}$ contains the eigenvectors and $\Lambda \in \mathbb{C}^{n \times n}$ is a diagonal matrix containing the corresponding eigenvalues. Orvieto et al. (2023) showed that we can rewrite the linear recurrent equation as:

$$h_t = P^{-1}\Lambda P h_{t-1} + B x_t \implies P h_t = \Lambda P h_{t-1} + P B x_t.$$

By defining $\tilde{h} \doteq Ph$ and $\tilde{B} \doteq PB$, we can write the new recurrence equation as follows:

$$\tilde{h}_t = \Lambda \tilde{h}_{t-1} + \tilde{B} x_t.$$

Since $\Lambda$ is a diagonal matrix, we can utilize its diagonal $\lambda \doteq \mathrm{diag}(\Lambda)$. The recurrent equation can be further simplified as:

$$\tilde{h}_t = \lambda \circ \tilde{h}_{t-1} + \tilde{B} x_t \tag{7}$$
$$y_t = \Re[C\tilde{h}_t] + D x_t$$

which is referred to as an independent recurrent module (Zucchet et al. 2023) because each element in the new state does not depend on any interaction with the other elements. Some instantiation

of independent recurrent modules with different details and assumptions are LRU (Orvieto et al. 2023), Online LRU Zucchet et al. (2023), eLSTM (Irie et al. 2024), RTU (Elelimy et al. 2024), Hawk (De et al. 2024), HGRN (Qin et al. 2024), MinGRU/MinLSTM (Feng et al. 2024), and columnar networks (Javed et al. 2023). It is worth mentioning that, in our analysis, we have not made any assumptions so far other than the linearity of $f$ and $g$. Thus, the equation with a complex-valued diagonal recurrence matrix is representationally equivalent to the original equation with a real-valued dense recurrence matrix.

### G.1 OPTIMIZATION ISSUES WITH RECURRENT LEARNING

One issue with the product term $\prod_{i=j+1}^{t} \frac{\partial \boldsymbol{h}_i}{\partial \boldsymbol{h}_{i-1}}$ in Eq. 3 and Eq. 4 is that it can vanish if the magnitude of the eigenvalues are less than 1 and explode if they are greater than 1. T-BPTT is less sensitive to this issue than BPTT. Further efforts include gating mechanisms to prevent vanishing or exploding gradients from excessive multiplication, like LSTM (Hochreiter and Schmidhuber 1997) and GRU (Cho 2014). Other efforts also restricted the eigenvalues of the product matrix to always be close to but less than 1 (e.g., Arjovsky et al. 2016). Recently, it was noticed that if the recurrent unit is restricted to be linear, controlling its eigenvalues becomes much easier, and thus optimization becomes more efficient (Zucchet and Orvieto 2024), which is what powers modern large-scale recurrent learning methods (e.g., Gu and Dao 2024, De et al. 2024, Dao and Gu 2024).

### G.2 NECESSITY OF COMPLEX NUMBERS LEARNING

The nature of input signals can vary from discrete to continuous based on the application. For example, they can be discrete, like language text, or continuous, like audio. Empirically, prior research (Gu and Dao 2024) showed that the recurrent system benefits from having complex-valued states in cases where the input signal is continuous and with little to no gain in the case of the discrete input (e.g., language). Thus, the recent recurrent systems with language models usually assume real-valued state $\boldsymbol{h}_t \in \mathbb{R}^n, \forall t$. In this primer, we focus on the general case where the state is complex-valued.

### G.3 STABILITY OF RECURRENT LEARNING

In Eq. 7, any entry in $\tilde{\boldsymbol{h}}_t$ can increase without bound if its corresponding eigenvalue is greater than or equal to 1, which makes the system unstable. To maintain stability, $\boldsymbol{\lambda}$ entries are restricted to have a magnitude less than 1. If rectangular representation, $a + ib$, is used, then $\sqrt{a^2 + b^2} < 1$ maintains stability. If trigonometric representation, $r(\cos(\theta) + i\sin(\theta))$, or exponential representation, $re^{i\theta}$, are used, then $r < 1$ maintains stability (Zucchet and Orvieto 2024, Elelimy et al. 2024). Further, if an entry $i$ in $\boldsymbol{\lambda}$ has a magnitude close to one, $|\lambda_i| \approx 1$, this might cause instability if the input contribution, $\tilde{\boldsymbol{B}}, \boldsymbol{x}_t$ is large (Orvieto et al. 2023). Thus, we can dampen the contribution of the input proportionally to the eigenvalue magnitude via multiplication by $\sqrt{1 - |\boldsymbol{\lambda}|}$. The resultant recurrence equation becomes $\tilde{\boldsymbol{h}}_t = \boldsymbol{\lambda} \circ \tilde{\boldsymbol{h}}_{t-1} + \boldsymbol{\gamma} \circ \tilde{\boldsymbol{B}} \boldsymbol{x}_t$, where $\boldsymbol{\gamma} = \sqrt{1 - |\boldsymbol{\lambda}|}$. In the following, we skip these stability modifications to have simpler derivations.

### G.4 SEPARATING COMPLEX INTO REAL AND IMAGINARY COMPONENTS

The linear recurrent unit in Eq. 7 can be implemented and used with automatic differentiation libraries as shown in LRU (Orvieto et al. 2023) and online LRU (Zucchet et al. 2023). However, automatic differentiation with complex numbers in existing software libraries is tricky and might give unexpected results (Elelimy et al. 2024) due to lack of adoption and support. Thus, it is better to separate the complex numbers into their real and imaginary components so that we have real equations and imaginary equations. The values of both components are real-valued, and the automatic differentiation libraries can deal with them more easily.

Let us separate the complex recurrence equation into two equations: real and imaginary. We define $\tilde{\boldsymbol{h}}_t = \tilde{\boldsymbol{h}}_t^R + i\tilde{\boldsymbol{h}}_t^I$, where $\tilde{\boldsymbol{h}}_t^R$ is the real part of the state vector and $\tilde{\boldsymbol{h}}_t^I$ is the imaginary part. We do the same trick for $\boldsymbol{\lambda}$ and $\tilde{\boldsymbol{B}}$: $\boldsymbol{\lambda} = \boldsymbol{\lambda}^R + i\boldsymbol{\lambda}^I$ and $\tilde{\boldsymbol{B}} = \tilde{\boldsymbol{B}}^R + i\tilde{\boldsymbol{B}}^I$

The recurrence equation is written as:

$$\tilde{h}_t^R + i\tilde{h}_t^I = (\boldsymbol{\lambda}^R + i\boldsymbol{\lambda}^I) \circ (\tilde{h}_{t-1}^R + i\tilde{h}_{t-1}^I) + (\tilde{B}^R + i\tilde{B}^I)\boldsymbol{x}_t$$
$$= \boldsymbol{\lambda}^R \circ \tilde{h}_{t-1}^R + i\boldsymbol{\lambda}^R \circ \tilde{h}_{t-1}^I + i\boldsymbol{\lambda}^I \circ \tilde{h}_{t-1}^R - \boldsymbol{\lambda}^I \circ \tilde{h}_{t-1}^I + \tilde{B}^R\boldsymbol{x}_t + i\tilde{B}^I\boldsymbol{x}_t$$

Let us separate the real components from the imaginary ones:

$$\tilde{h}_t^R = \boldsymbol{\lambda}^R \circ \tilde{h}_{t-1}^R - \boldsymbol{\lambda}^I \circ \tilde{h}_{t-1}^I + \tilde{B}^R\boldsymbol{x}_t$$
$$i\tilde{h}_t^I = i\boldsymbol{\lambda}^R \circ \tilde{h}_{t-1}^I + i\boldsymbol{\lambda}^I \circ \tilde{h}_{t-1}^R + i\tilde{B}^I\boldsymbol{x}_t$$

Note how we can drop $i$ from both sides of the imaginary equations and retain a real-valued equation.

The recurrent state of such a separated system can be seen as $\tilde{h}_t^{\text{combined}} = [\tilde{h}_t^R; \tilde{h}_t^I] \in \mathbb{R}^{2n}$. One advantage of such a view is that we no longer need to learn a complex-valued $\boldsymbol{C}$ matrix. The output $\boldsymbol{y}_t$ is given by $\boldsymbol{y}_t^{\text{combined}} = \tilde{h}_t^{\text{combined}}\boldsymbol{C}^{\text{combined}}$, where $\boldsymbol{C}^{\text{combined}} \in \mathbb{R}^{m \times 2n}$. This is representationally equivalent to learning a complex-valued state $\tilde{h}_t \in \mathbb{R}^n$ and $\boldsymbol{C} \in \mathbb{C}^{m \times n}$ where the output is given as $\boldsymbol{y}_t = \Re[\boldsymbol{C}\tilde{h}_t]$. This is because $\boldsymbol{y}^{\text{combined}} = \begin{bmatrix} \tilde{h}_t^R \\ \tilde{h}_t^I \end{bmatrix}[\boldsymbol{C}_1 \ \boldsymbol{C}_2] = \boldsymbol{C}_1\tilde{h}_t^R + \boldsymbol{C}_2\tilde{h}_t^I$, where $\boldsymbol{C}_1, \boldsymbol{C}_2 \in \mathbb{R}^{m \times n}$. On the other hand, $\boldsymbol{y}_t = \Re[\boldsymbol{C}\tilde{h}_t] = \Re[(\boldsymbol{C}_R + i\boldsymbol{C}_I)(\tilde{h}_t^R + i\tilde{h}_t^I)] = \boldsymbol{C}_R\tilde{h}_t^R - \boldsymbol{C}_I\tilde{h}_t^I$. If we set $\boldsymbol{C}_R = \boldsymbol{C}_1$ and $\boldsymbol{C}_I = -\boldsymbol{C}_2$, then we get $\boldsymbol{y}_t = \boldsymbol{y}_t^{\text{combined}}$.

The cosine representation is better than the rectangular representation for learning (Elelimy et al. 2024) since we can directly restrict the magnitude of $\boldsymbol{\Lambda}$ eigenvalues. Utilizing the cosine representation of complex numbers: $a + ib = r\cos(\theta) + i\sin(\theta)$, we can write the two equations differently. We define $\boldsymbol{\lambda}^R = \boldsymbol{r}\cos(\boldsymbol{\theta})$, $\boldsymbol{\lambda}^I = \boldsymbol{r}\sin(\boldsymbol{\theta})$. The resultant recurrence equations are given by:

$$\tilde{h}_t^R = \boldsymbol{r} \circ \cos(\boldsymbol{\theta}) \circ \tilde{h}_{t-1}^R - \boldsymbol{r} \circ \sin(\boldsymbol{\theta}) \circ \tilde{h}_{t-1}^I + \tilde{B}^R\boldsymbol{x}_t,$$
$$\tilde{h}_t^I = \boldsymbol{r} \circ \cos(\boldsymbol{\theta}) \circ \tilde{h}_{t-1}^I + \boldsymbol{r} \circ \sin(\boldsymbol{\theta}) \circ \tilde{h}_{t-1}^R + \tilde{B}^I\boldsymbol{x}_t,$$

which recovers RTU unit (Elelimy et al. 2024).

### G.5 ENFORCING CONJUGATE PAIRS LEARNING

Since the recurrent unit matrices are real, $\boldsymbol{A} \in \mathbb{R}^{n \times n}$, $\boldsymbol{B} \in \mathbb{R}^{n \times d}$, and $\boldsymbol{C} \in \mathbb{R}^{n \times m}$, then $\boldsymbol{\Lambda}$ must contain complex conjugate pairs. It might be beneficial to enforce conjugate pairs in the learning process. We tie the weights of $\boldsymbol{\Lambda} \in \mathbb{C}^{n \times n}$ enforce conjugate pairs and write it as

$$\boldsymbol{\Lambda}_{\text{restricted}} = \begin{bmatrix} \lambda_1 & 0 & 0 & \cdots & 0 \\ 0 & \lambda_2 & 0 & \cdots & 0 \\ \vdots & \vdots & \ddots & \cdots & \vdots \\ 0 & 0 & 0 & \lambda_{2n-1}^* & 0 \\ 0 & 0 & 0 & \cdots & \lambda_{2n}^* \end{bmatrix} = \begin{bmatrix} \boldsymbol{\Lambda}_{\text{original}} \\ \boldsymbol{0} \end{bmatrix} + \begin{bmatrix} \boldsymbol{0} \\ \boldsymbol{\Lambda}_{\text{conjugate}} \end{bmatrix},$$

where $\boldsymbol{\Lambda}_{\text{original}}, \boldsymbol{\Lambda}_{\text{conjugate}} \in \mathbb{C}^{\frac{n}{2} \times \frac{n}{2}}$. Here, $\lambda_k, \lambda_k^*, \forall k$ are conjugate pairs representing complex eigenvalues. We call recurrent units with this restriction as restricted complex-valued recurrence.

### G.6 LEARNING WITH RTRL

To be able to learn with RTRL and the complex recurrent unit, $\tilde{h}_t = \boldsymbol{\lambda} \circ \tilde{h}_{t-1} + \tilde{B}\boldsymbol{x}_t$, we need to compute the sensitivity matrices for $\boldsymbol{\lambda}$, and $\tilde{B}$. We denote $\mathbf{S}^{\boldsymbol{\lambda}} \doteq \frac{\partial h_t}{\partial \boldsymbol{\lambda}}$ and $\mathbf{S}^{\tilde{B}} \doteq \frac{\partial h_t}{\partial \tilde{B}}$ to the sensitivity matrices for $\boldsymbol{\lambda}$, and $\tilde{B}$, respectively. We refer the reader to Zucchet et al. (2023) for full derivation and analysis for RTRL for complex-valued LRU. In addition, we refer the reader to Elelimy et al. (2024) for a derivation for complex-based LRU using a real-valued system. Here, we provide the derivation for completeness. The sensitivity matrices update equation is given as follows:

$$S_{t,i,j}^{\boldsymbol{\lambda}} = \frac{\partial h_{t,i}}{\partial \lambda_j} = \frac{\partial}{\partial \lambda_j}\left(\lambda_i\tilde{h}_{t-1,i} + \sum_m \tilde{B}_{i,m}x_{t,m}\right)$$

$$= \delta_{i,j}\tilde{h}_{t-1,i} + \delta_{i,j}\lambda_i S^{\boldsymbol{\lambda}}_{t-1,i,j}$$

$$S^{\tilde{\boldsymbol{B}}}_{t,i,j,k} = \frac{\partial h_{t,i}}{\partial B_{j,k}} = \frac{\partial}{\partial B_{j,k}}\left(\lambda_i \tilde{h}_{t-1,i} + \sum_m \tilde{B}_{i,m} x_{t,m}\right)$$

$$= \lambda_i S^{\tilde{\boldsymbol{B}}}_{t-1,i,j,k}\delta_{i,j} + \sum_m \delta_{i,j}\delta_{k,m}x_{t,m}$$

$$= \lambda_i S^{\tilde{\boldsymbol{B}}}_{t-1,i,j,k}\delta_{i,j} + \delta_{i,j}x_{t,k}.$$

We can also write the recursive relationship using the reduced sensitivity objects as follows:

$$\mathbf{S}^{\boldsymbol{\lambda}}_t = \frac{\partial}{\partial\boldsymbol{\lambda}}\left(\boldsymbol{\lambda}\circ\tilde{\boldsymbol{h}}_{t-1} + \tilde{\boldsymbol{B}}\boldsymbol{x}_t\right) = \boldsymbol{\lambda}\circ\mathbf{S}^{\boldsymbol{\lambda}}_{t-1} + \tilde{\boldsymbol{h}}_{t-1}$$

$$\mathbf{S}^{\tilde{\boldsymbol{B}}}_t = \frac{\partial}{\partial\tilde{\boldsymbol{B}}}\left(\boldsymbol{\lambda}\circ\tilde{\boldsymbol{h}}_{t-1} + \tilde{\boldsymbol{B}}\boldsymbol{x}_t\right) = \text{Diag}(\boldsymbol{\lambda})\mathbf{S}^{\tilde{\boldsymbol{B}}}_{t-1} + \mathbf{1}\boldsymbol{x}_t^\top$$

where $\circ$ denotes element-wise product. Note how the matrix $\mathbf{S}^{\boldsymbol{\lambda}}_t$ reduces to a vector since $\delta_{i,j} = 0, \forall i \neq j$. Similarly, the 3d tensor $\mathbf{S}^{\tilde{\boldsymbol{B}}}_t$ reduces to a 2d matrix. This structure in the sensitivity objects is a result of the structure of the independent recurrent module.

## H  PRIMER ON PARALLEL SCAN

Parallel scan (Blelloch 1990) is an operation that applies a binary associative operator $\bullet$ on a number of elements $L$ in a certain way. Let us consider the linear recurrence $\boldsymbol{h}_{k+1} = \boldsymbol{A}_k\boldsymbol{h}_k + \boldsymbol{B}_k\boldsymbol{x}_k$. For a sequence of length $L$, we can write the elements belonging to each step $k$ as $c_k = (\boldsymbol{A}_k, \boldsymbol{B}_k\boldsymbol{x}_k)$. The elements $\{c_1, \ldots, c_L\}$ are precomputed before applying the parallel scan operator. The binary associative operator $\bullet$ of this recurrence is given by $q_i \bullet q_j \doteq (q_{j,1}\Box q_{i,1}, q_{j,1}\Diamond q_{i,2} + q_{k,2})$, where $q_{i,1}$ is the 1st entry of the $i$th element, $q_{i,2}$ is the 2nd entry of the $i$th element, $\Box$ denotes matrix-matrix multiplication, $\Diamond$ denotes matrix-vector multiplication, and $+$ denotes element-wise addition.

First, we perform the *upsweep*, where we recursively combine adjacent pairs of elements to build a binary tree. At the bottom level, we combine $(c_1, c_2), (c_3, c_4), \ldots, (c_{L-1}, c_L)$. Each pair is combined using $\bullet$, and the resulting values form the next level. This process continues until we reach the root. For example, a node covering $c_3, c_4, c_5, c_6$ will store $c_3 \bullet c_4 \bullet c_5 \bullet c_6$. These values are reused in the next phase. Second, we perform the *downsweep* to turn these tree values into the actual cumulative products. Starting from the root node then traversing the tree, we pass the value of the node to its children where the right child gets the value of the parent combined with the left child's value and the left child gets the same value as the parent. After this process is complete, each leaf node $k$ contains the cumulative product $s_k = c_1 \bullet \cdots \bullet c_{k-1}$, containing all hidden states $\{h_1, \ldots, h_L\}$.

Because both upsweep and downsweep take $O(\log L)$ depth (by combining or distributing pairs in parallel), all hidden states are produced in $O(\log L)$ parallel time. The computational complexity is $O(M \log L)$, where $M$ is the cost of matrix-matrix multiplication. Specifically, the cost is $O(n^3 \log L)$ using dense recurrence matrix $\boldsymbol{A} \in \mathbb{R}^{n \times n}$ and $O(n \log L)$ for diagonal $\boldsymbol{A}$.

