# OpenReview forum: "Sequence Learning from Continuous Streams of Data"
_ICLR.cc/2026/Conference — ICLR 2026 Conference Withdrawn Submission_

### Official Review · Reviewer_Sv1G · 2025-10-24

**Soundness:** 3
**Presentation:** 3
**Contribution:** 2
**Rating:** 4
**Confidence:** 3

**Summary:**

The paper addresses learning from continuous data streams by formalizing multi-stream sequence learning, where training proceeds in blocks but the model’s hidden state persists across updates and is reset only at episode/document boundaries (unlike IID chunking). To exploit this regime, the authors propose Memora, a recurrent-only architecture built on a Gated Linear Recurrent Unit (GLRU) that supports both parallel T-BPTT via scan and true online learning via RTRL. Across selective copying, byte-level language modeling and DNA modeling, the approach shows strong performance.

**Strengths:**

The paper is clearly written and easy to read.

The experiment suite is very broad evaluating the proposed approach across many domains and tasks.

**Weaknesses:**

I think my main concern is that the components the authors describe here have been present in literature. For instance, multi-stream learning is not new and was adopted in TransformersXL (https://arxiv.org/abs/1901.02860). However, this is okay since they don't claim they introduce this framework, they mainly formalize it. However, for the memora architecture, the idea of gated recurrence has been known in literature for a long time with LSTM and GRU networks. I am curious how memora differs from them (apart from the formulation)?

From the language modelling results, this approach seems to help at low sequence lengths (8-32) but matches performance (or performs worse?) of transformers at high sequence lengths (512-1024). This reduces the effectiveness of the approach as at scale ideally one may want an approach that retains performance gain as we scale sequence length as at scale we generally don't train models at <100 sequence length.

I think there should be a transformerXL baseline for the multi-stream setting. Similarly, even for Mamba and Hawk in the multi-stream setting, have the authors provided a state which persists across sequences? Also there are many approaches to maintain context for transformers beyond the current context window such as sink attention (https://arxiv.org/abs/2309.17453). I believe the authors should compare against either one of these baselines.

**Questions:**

Is there a reason why the authors chose to explore language modelling at byte level and not token level?

---

### Official Review · Reviewer_AUJY · 2025-10-28

**Soundness:** 3
**Presentation:** 3
**Contribution:** 3
**Rating:** 6
**Confidence:** 2

**Summary:**

The paper proposes Memora, a recurrent-only sequence model built on the Gated Linear Recurrent Unit (GLRU), and introduces a multi-stream training paradigm that preserves temporal continuity instead of using IID-chunked data. By keeping hidden states persistent across updates, the method enables more effective long-range learning and supports both truncated BPTT and efficient online RTRL training.

**Strengths:**

1. The architecture of Memora is simple with GLRU being linear in state, enabling stable long-sequence learning without complex state-space or attention mechanisms.
2. The paper highlights how IID chunking breaks temporal continuity, which is a widely overlooked issue in current sequence model training.
3. The experiments demonstrate consistent gains on byte-level text and DNA tasks, showing Memora works in a strict streaming setting.

**Weaknesses:**

1. While multi-stream training improves continuity and performance, it likely introduces sequential dependencies that reduce parallelism compared to IID chunking. Without throughput or wall-clock benchmarks, it is unclear whether the exact efficiency of this approach.
2. Gradient stability is discussed intuitively but not proven or quantitatively measured.

**Questions:**

1. What is the memory footprint of maintaining persistent states across streams?
2. Does performance still improve if you partially reset hidden states (e.g., at paragraph/section boundaries)?

---

### Official Review · Reviewer_oyzC · 2025-10-31

**Soundness:** 2
**Presentation:** 2
**Contribution:** 1
**Rating:** 4
**Confidence:** 3

**Summary:**

The paper formalizes a multi-stream sequence learning setting where several continuous streams are trained jointly without resetting hidden states at chunk boundaries. This contrasts with IID chunking and allows models with persistent state (recurrent or state-space) to learn longer dependencies. Multi-stream training exploits this persistent state: recurrent models that carry state across contiguous blocks recover long-range dependencies even with tiny strides (e.g., S = 2), whereas Transformers (e.g., LLaMA) don’t benefit because their context resets each update. In other words, it doesn’t matter that the previous chapter of a book was in the last batch and the current one is in this batch. Transformers cannot pass state across batches.

The paper proposes the GLRU unit, designed so that: 1) T-BPTT (truncated backprop through time) + scan (Blelloch, 1990) enable efficient, parallel training on long sequences. 2) RTRL (real-time recurrent learning with T = 1) enables true online “one-step-at-a-time” learning with many parallel streams. The model’s diagonal sensitivities make RTRL tractable.

Table 1 compares GLRU with LRU, RG-LRU, and GRU, showing that only GLRU and RG-LRU permit both efficient scan and RTRL, while GLRU additionally allows state expansion (mapping inputs to a higher-dimensional state via a matrix multiplication).

**Strengths:**

- The paper’s multi-stream formulation isolates a training regime (continuous, non-IID streams) that aligns closely with realistic continual-learning scenarios, though this is not the setting mainly explored in the paper.
- The theoretical derivations for both scan-based T-BPTT and online RTRL in diagonally gated units are clean and internally consistent (as far as I could check). GLRU and RG-LRU share this mathematical tractability.
- The empirical section shows that multi-stream training benefits recurrent/state-space models, while standard Transformers show no gain under the same protocol—supporting the main premise.

**Weaknesses:**

- The architectural novelty of GLRU over RG-LRU appears minimal: primarily the inclusion of a state-expansion matrix $B$ (a single linear projection $B x_t$ ) and a slightly altered gating parameterization. This expansion allows the recurrent state dimension $n > d$ , but mechanistically it’s just a matrix multiplication before elementwise gating.
- The paper emphasizes this as a key differentiator (in Table 1 for instance), yet I couldn't find a direct empirical ablation between GLRU and RG-LRU. As far as I can tell, the numerical results compare Memora(+GLRU) to Hawk (LRU-based), LRU, MinGRU, and Transformers—not to RG-LRU.
  Hence, the claimed improvement from $B x_t$ remains unverified experimentally.
- Without a head-to-head GLRU–to–RG-LRU comparison under identical conditions (same parameter budget, data, optimizer, etc.), the true benefit of $B x_t$ and the new gating design remains ambiguous.

**Questions:**

- Eventually, whole books or documents can fit into long-context Transformer models, as their context length can now reach hundreds of thousands of tokens. If the end goal is efficient sequence modeling (good performance per FLOP), how does this approach compare? The main competitor—long-context Transformers—is not directly compared against.
- In equation (6), since the recurrence mixes $h_{t-1}$ elementwise (Hadamard), the Jacobians $\partial h_t / \partial h_{t-1}$ are diagonal, which makes parallel scan possible for T-BPTT and yields stable, efficient recursions. Do I understand this correctly?
- Is the improvement over RG-LRU truly limited to the state-expansion term $B x_t$​ ? If so, could RG-LRU not trivially match GLRU by adding a learned input projection layer? Do you think the empirical advantage persist after such an equalized comparison?
- The experiments compare models with roughly matched parameter counts and training data exposure (iterations or bytes), but not under matched FLOPs or wall-clock time. Is there any comparison in actual training cost or throughput (e.g., tokens per second, total FLOPs, or wall clock time) against Transformers for instance?

---

### Official Review · Reviewer_aQKW · 2025-11-01

**Soundness:** 2
**Presentation:** 3
**Contribution:** 2
**Rating:** 4
**Confidence:** 3

**Summary:**

This work addresses the issue of missing long-distance dependencies that arise when training language models on separated text chunks. By introducing a Gated Linear Recurrent Unit (GLRU) with parallel streams, the proposed method achieves improved performance on sequential modeling tasks.

**Strengths:**

1. The writing and presentation are clear and logically consistent.
2. Comprehensive quantitative experiments are conducted to demonstrate the effectiveness of the proposed method.

**Weaknesses:**

1. As stated in line 53, this work is inspired by [1], which addresses the problem of learning long-distance dependencies in Transformers. However, the discussion comparing the proposed method with [1] is missing. To strengthen the soundness, it would be better to clarify in what aspects Memora outperforms Transformer-XL, and to provide reasoning or evidence supporting these claims.
2. While Transformers are known for their efficiency in parallel processing across the context during training, the main task in this study focuses on sequential modeling. Therefore, the efficiency of different architectures should also be considered and discussed.


[1]. Transformer-XL: Attentive Language Models Beyond a Fixed-Length Context.

**Questions:**

Gradient explosion and vanishing are well-known issues in recurrent models. Without truncating gradients through time, optimization may become unstable; however, truncation can also result in the loss of long-distance dependencies. How does the proposed method address these challenges?

---

### Note · Authors · 2025-12-03

**Comment:**

We request the withdrawal of our submission “Sequence Learning from Continuous Streams of Data” (Submission number: 22149) from consideration for ICLR 2026. We thank the reviewers and AC for their time and effort.

**Withdrawal Confirmation:**

I have read and agree with the venue's withdrawal policy on behalf of myself and my co-authors.